# A Game-theoretic Approach to Personalized Federated Learning Based on Target Interpolation

## Abstract

Contrary to classical Federated Learning (FL) that focuses on collaborative learning of a shared global model via a central server, Personalized Federated Learning (PFL) trains a separate model for each user in order to address data heterogeneity and meet local demands. This paper proposes `pFedGT`, a method for personalized Federated Learning based on a Game-theoretic approach, that adopts a novel formulation termed "Target interpolation." In specific, each user solves a local optimization problem that comprises of a weighted average of two terms: one for the local loss (based on the user's data) and one for the global loss (based on all the data in the system). The latter is, of course, not accessible to the users (due to the large data volumes and privacy concerns) and it is approximated using second-order expansion which allows for an efficient federated implementation. In `pFedGT`, the users play a game (by minimizing their local problems), and the algorithm supports partial participation in each round. We prove existence and uniqueness of a *Nash equilibrium* and establish a linear convergence rate under standard assumptions. Extensive experiments on real datasets under variable levels of statistical heterogeneity are used to portray the merits of the proposed solution. In particular, our method achieves on average $2.6\%$ and $3.0\%$ higher accuracy on CIFAR-10 and CIFAR-100 datasets, and $3.17\%$ on HAR dataset than leading baselines.

## 1 Introduction

The proliferation of mobile phones, wearable devices, and autonomous vehicles has resulted in a substantial surge in the generation of distributed data. Distributed machine learning (ML) techniques (Bottou, 2010; Dean et al., 2012) facilitate the seamless integration of artificial intelligence into the Internet of Things (IoT). In this setting, Federated Learning (FL) (Konečný et al., 2016; McMahan et al., 2017) has emerged as a novel paradigm enabling model training on edge devices using local data and communication with a server, while simultaneously prioritizing data privacy and minimizing communication overhead. Numerous FL methodologies, such as `FedDyn` (Durmus et al., 2021), `FedVARP` (Jhunjhunwala et al., 2022), and `FedExP` (Jhunjhunwala et al., 2023), as well as primal-dual methods like `FedPD` (Zhang et al., 2021), `FedADMM` (Gong et al., 2022; Wang et al., 2022), and `FedHybrid` (Niu & Wei, 2023), have been proposed to tackle challenges associated with non-independently and non-identically distributed (non-IID) data.

With the growing emphasis on personalized services, personalization is emerging as a prominent technique aimed at training models customized to fulfill the specific needs of individual users. To that end, a key challenge is the possibly limited data volume on the user side, which may result in poor generalization of locally trained models. This can be remedied by transfer learning, i.e., intend to boost personal models through federated communication exchanges. However, all the aforementioned methods aim to learn a single global model, which can not suffice to explain the data of individual users and meet their personalized demands.

Personalized Federated Learning (PFL) differs from conventional FL in that it seeks to train multiple models tailored to meet the particular demands of individual users. This approach (Kulkarni et al., 2020; Tan et al., 2022) proves particularly beneficial when dealing with users who exhibit varying

levels of data distribution across the system. There has been an increasing number of methods developed to address the challenges associated with conventional FL, and to further improve the accuracy, efficiency, and data privacy protection of personalized models (see Sec. 2).

In this paper, we propose a game-theoretic-based approach method for PFL, termed `pFedGT`. To address statistical heterogeneity and accommodate diverse user demands, we introduce a novel domain-based approach called "target interpolation." This enables users to capture both local and global preferences by reformulating the user's objective function as a weighted average of local and global loss. Given stringent privacy concerns and communication overhead, it is impractical for users to compute the global loss (since data exchanges are out of the question). To tackle this issue, we employ a quadratic approximation based on the second-order Taylor expansion with respect to their local models. Furthermore, to reduce communication overhead, we combine communication messages on the user side. Consequently, we formulate the PFL framework into a scenario where users engage in a non-cooperative game since they are selfish in solving their local reformulated problems without any consideration of the impact they have on others. We allow partial user participation in the operation of the algorithm, and rigorously establish its convergence.

**Contributions**:
- We propose a new domain-based approach, termed "target interpolation," to model user collaboration on the domain field. This is accomplished by reformulating the user's local objective function to a weighted average of local and global loss.
- We propose a novel game-theoretic-based PFL method, termed `pFedGT`. Based on "target interpolation," `pFedGT` approximates the global loss by using a second-order expansion centered at the user model (in consideration of communication overhead and privacy concerns). Afterwards, users interact with each other selfishly, prioritizing the minimization of their individually reformulated loss without considering the impact on others. Furthermore, the server employs a tracking average of all users' updated messages, enabling any partial participation schemes in deploying our method.
- We demonstrate the existence and uniqueness of a *Nash equilibrium* under standard assumptions. Additionally, we establish a linear convergence rate for our proposed algorithm in a general nonconvex setting.
- We have conducted numerous experiments on federated neural network training on real datasets in various non-IID settings. Our experiments demonstrate that `pFedGT` outperforms state-of-the-art methods, achieving an average accuracy improvement of $2.6\%$ on CIFAR-10, $3.0\%$ on CIFAR-100, and $3.17\%$ on the HAR dataset.

## 2 RELATED WORK

### 2.1 PERSONALIZED FEDERATED LEARNING

Numerous techniques have been proposed for PFL. (Smith et al., 2017) proposed `MOCHA`, a multi-task method that allows discovering commonalities with other users using alternating minimization (on regularized loss and weights of regularization); however, the proposed primal-dual method is limited in the convex setting. Another popular approach is inspired by model agnostic meta-learning (`MAML`) (Finn et al., 2017), which builds the global model based on multiple tasks. Subsequently, users personalize their local models based on the global model. Nevertheless, the `MAML` framework requires high computation and storage costs since it involves second-order information (Hessian matrix); this was handled in (Nichol et al., 2018) via approximation only using first-order derivatives. Another line of methodology combines the global and local models via weighted averaging (Grimberg et al., 2021; Deng et al., 2020; Hanzely & Richtárik, 2020).

There have also been methods that leverage the neural network architecture, as inspired by the concept of representation learning (Bengio et al., 2013). (Arivazhagan et al., 2019; Collins et al., 2021; Oh et al., 2022) proposed to train the base layers of the network using standard FL, while users update their top layers (personalization layers) by means of local training. Other approaches to training global and local models involve using different regularization techniques, as in (Huang et al., 2021) and (T Dinh et al., 2020). The regularization terms can help to decouple personalized and global models by combining a coefficient to control the difference between them. Another approach involves clustering methods, where a separate model is trained for each user cluster, assuming that

the local data of each client may share the same context. This allows a group of users to learn a group model, as shown in (Mansour et al., 2017; Ghosh et al., 2020; Sattler et al., 2020).

Unlike previous approaches that utilize weighted averaging of global and local models, `pFedGT` interpolates the local objective function with the global loss, which enables a more direct incorporation of both local and global preferences. Moreover, `pFedGT` is model-agnostic, in that it does not require specific modifications for different layers of the network.

## 2.2 GAME-THEORETIC APPROACHES IN FEDERATED LEARNING

Research efforts have been dedicated to developing incentive mechanisms (Tu et al., 2022; Zeng et al., 2022; Wang et al., 2023) aimed at consistently enhancing the performance of the global model. (Pandey et al., 2020) model federated learning as a Stackelberg game, subsequently, they reformulate the FL problem as a utility maximization problem. Additionally, (Song et al., 2019) utilize the Shapley value (SV) to quantify each user's data contribution and use this to proportionally compensate users so as to continually attract high-quality participants. Similar to (Song et al., 2019), (Wu et al., 2022) also utilize the SV to quantify users' marginal contribution while they also capture the effect of collaboration in achieving personalization.

The application of game theory in FL is quite natural, as users interact with each other to obtain benefits rather than engaging solely in local training with a limited amount of private data. However, existing game-theoretic FL approaches have introduced a significant level of complexity (i.e., compute the SV) compared to classical FL algorithms.

## 3 ALGORITHM

### 3.1 PROBLEM FORMULATION

To accommodate statistical heterogeneity, PFL strives to harness the private training datasets of individual users for the collaborative training of personalized models $(w_1, \ldots, w_m)$, where $m$ represents the total number of users. These personalized models are engineered to deliver superior local performance compared to both the global model and models trained independently.

In this paper, we propose a domain-based approach to address personalization in FL, which we term "target interpolation." This method involves directly interpolating the local objective with the global target, which is accomplished by formulating the local problem as minimizing a weighted average of the local loss and global loss. In specific, the local problem for user $i$ is as follows:

$$\underset{w_i \in \mathbb{R}^d}{\text{minimize}} \; \gamma_i f_i(w_i) + (1 - \gamma_i) \frac{1}{m} \sum_{j=1}^{m} f_j(w_i), \tag{1}$$

where $\gamma_i \in [0, 1]$ is the personalization coefficient. Note that when $\gamma_i = 1$, this corresponds to a local training paradigm that solely relies on local data. On the other hand, when $\gamma_i = 0$, it represents conventional FL with no personalization.

To ascertain the feasibility of our proposed methodology, we conducted a battery of preliminary experiments on a single user based on formulation (1). The exchange of data between users would be a necessary step for solving (1), which is **not** feasible and or permissible in FL. The influence of varying degrees of personalization on model performance can be explicitly observed in Fig. 1. Our experimental results illustrate the feasibility of the proposed formulation while shedding light on the impact of statistical heterogeneity on the choice of personalization hyperparameter $\gamma_i$.

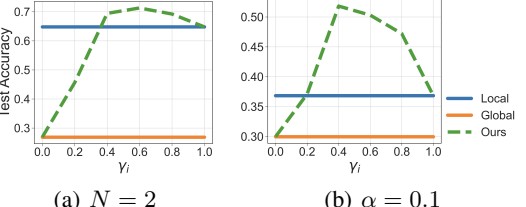

(a) $N = 2$        (b) $\alpha = 0.1$

Figure 1: The effect of the personalization hyperparameter $\gamma_i$ for a single user based on the formulation presented in (1). The experiments were performed on the CIFAR-10 dataset, with $N$ and $\alpha$ representing the level of statistical heterogeneity (see Sec. 5). In both cases, target interpolation attains higher accuracy than both local training and classical FL.

To tackle the issue of unavailability of the second sum-term in (1) at the user side, we use a quadratic approximation as:

$$f_j(w_i) \approx f_j(w_j) + \nabla f_j(w_j)^\top (w_i - w_j) + \frac{\mu}{2} \|w_i - w_j\|^2.$$

This is an approximate second-order Taylor expansion that ignores the specific Hessian dependency, due to the large model size and communication overhead.

Furthermore, we introduce a regularization term into the local problem aiming at reducing model complexity and mitigating the risk of overfitting. In the end, the local problem for user $i$ is to

$$
\begin{aligned}
&\underset{w_i \in \mathbb{R}^d}{\text{minimize}} && G_i(w_i; w_{-i}) := F_i(w_i; w_{-i}) + \frac{\rho}{2} \|w_i\|^2, \\
&\text{where } F_i(w_i; w_{-i}) := \gamma_i f_i(w_i) + \frac{1 - \gamma_i}{m} \sum_{j=1}^m \left( f_j(w_j) + \nabla f_j(w_j)^\top (w_i - w_j) + \frac{\mu}{2} \|w_i - w_j\|^2 \right), \\
&&& w_{-i} := (w_1, \ldots, w_{i-1}, w_{i+1}, \ldots, w_m).
\end{aligned}
\tag{2}
$$

Each user in the system is selfish, solely dedicated to maximizing its local model fidelity. Consequently, the PFL problem is cast into a non-cooperative game, where each user solves problem (2) (using information obtained through server aggregation) so that the system converges to a *Nash equilibrium* defined as:

$$G_i(w_i^\star; w_{-i}^\star) \leq G_i(w_i; w_{-i}^\star), \forall w_i, \forall i. \tag{3}$$

## 3.2 Proposed Algorithm: pFedGT

In `pFedFT`, each participating user within the system solves its corresponding (2) (in parallel and inexactly). We employ stochastic gradient descent (SGD) (for ease of notation we show the full gradient here) to solve the corresponding minimization problem. At the global round $t \in \{1, \ldots, T\}$ and local epoch $e \in \{1, \ldots E\}$, the gradient of $G_i(\cdot)$ over local model $w_i$ is as follows:

$$\nabla_{w_i} G_i(w_i^{t,e}; w_{-i}^t) = \gamma_i \nabla f_i(w_i^{t,e}) + \frac{1 - \gamma_i}{m} \nabla f_i(w_i^{t,e}) + \frac{1 - \gamma_i}{m} \sum_{j \neq i}^m \left( \nabla f_j(w_j^t) + \mu(w_i^{t,e} - w_j^t) \right) + \rho w_i^{t,e}$$

$$= \gamma_i \nabla f_i(w_i^{t,e}) + \frac{1 - \gamma_i}{m} \nabla f_i(w_i^{t,e}) + \frac{(1 - \gamma_i)(m-1)}{m} \mu w_i^{t,e} + \rho w_i^{t,e} + \frac{1 - \gamma_i}{m} \sum_{j \neq i}^m \left( \nabla f_j(w_j^t) - \mu w_j^t \right). \tag{4}$$

It is important to emphasize that in the augmented local problem (2), user $i$ only updates $w_i^{t,e}$, while terms related to the model/gradients of other users ($w_j^t$) reflect previous values. This is amenable to a federated implementation by noting that these can be grouped into a sum term in the equation above; this can be obtained by server aggregation (since the missing term in the sum, i.e., $j = i$ is locally available).

To reduce the communication overhead, users exchange $c_i := \nabla f_i(w_i) - \mu w_i, \forall i$, which is updated as Alg. 2, step 6. In view of partial participation ($S^t$ represents the set of participants at round $t$), the aggregation can be carried (see Alg. 1 step 7), by tracking average as:

$$c^{t+1} = c^t + \frac{\lambda}{|S^t|} \sum_{i \in S^t} (c_i^{t+1} - c_i^t). \tag{5}$$

It is not hard to verify that if $\lambda = |S^t|/m$ this computes the true average $\frac{1}{m} \sum_{i=1}^m c_i$ (since for non-participating users $c_i^{t+1} \equiv c_i^t$) (see Lemma 1 in Appendix B). However, in the proposed method we allow for a more general choice of $\lambda$ (i.e., a generalized step size parameter), which serves to balance the influence of past information and information pertaining to the current round, in order to robustify the algorithm in a practical setting (e.g., reduce oscillations due to heterogeneity).

In practical scenarios, it is valuable for the cloud server to uphold a global model as a meta-model. In our proposed algorithm, alongside the average of local combined messages, the server also maintains a server model denoted as $\theta$, which is updated using tracking aggregation:

$$\theta^{t+1} = \theta^t + \frac{\eta_s}{|S^t|} \sum_{i \in S^t} (w_i^{t+1} - w_i^t), \tag{6}$$

---

**Algorithm 1** `pFedGT`

---

**Input**: Total number of rounds $T$, aggregation step size $\lambda$, and server model step size $\eta_s$.

1: **for** $t = 0, 1, \ldots, T - 1$ **do**
2:     Server selects $S^t \subset [m]$
    Clients:   *// In parallel*
3:     **for** $i \in S^t$ **do**
4:        download $\theta^t, c^t$ from the server
5:        $(w_i^{t+1}, c_i^{t+1}) \leftarrow \texttt{ClientUpdate}(\theta^t, c^t)$
6:     **end for**
    Server:
7:     $c^{t+1} = c^t + \frac{\lambda}{|S^t|}\sum_{i \in S^t}(c_i^{t+1} - c_i^t)$
8:     $\theta^{t+1} = \theta^t + \frac{\eta_s}{|S^t|}\sum_{i \in S^t}(w_i^{t+1} - w_i^t)$
9: **end for**

---

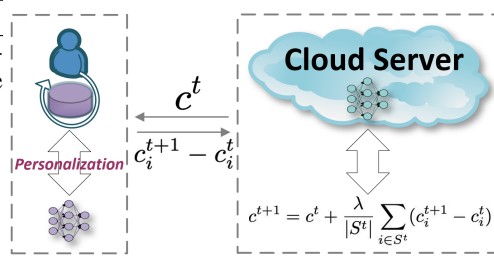

Figure 2: Architecture of `pFedGT`. At global round $t$, active user $i$ downloads the aggregated messages of all users $c^t$ from the cloud server. Following this, user $i$ solves (2) using its local private data. Subsequently, user $i$ uploads the difference of combined messages $c_i^{t+1} - c_i^t$ to the cloud server, which aggregates them.

---

**Algorithm 2** ClientUpdate($\theta^t, c^t$)

---

**Input**: Local epoch number $E$, client learning rate $\eta$, hyperparameters $\gamma_i, \mu, \rho$.

1: Initialize $w_i, c_i$
2: **for** $e = 0, 1, \ldots, E - 1$ **do**
3:     **for** each batch $b$ **do**
4:        Compute batch gradient $\nabla f_i(w_i; b)$
5:        $w_i \leftarrow w_i - \eta\big(\nabla f_i(w_i; b) + (1 - \gamma_i)(c^t - c_i) + (1 - \gamma_i)\frac{1}{m}(c_i - c_i^t) + \rho w_i\big)$
6:        $c_i \leftarrow \nabla f_i(w_i; b) - \mu w_i$
7:     **end for**
8: **end for**
9: Return $w_i, c_i$

---

where $\eta_s$ is the server model step size. Given that we can establish strong convexity of problem (2) under Assumption 1 (despite the non-convexity of the loss functions), we employ this server model as the initialization in (modified) local training (line 1 in Alg. 2). Our experimental results (Fig. 5 and Fig. 6) demonstrate that this server model $\theta$ can not only serve as a good choice for warmstarting the client update process (Alg. 2) but also as a well-suited pre-trained model at the system-level. Besides, we choose to initialize the local $c_i$ with $c$ before local training to avoid oscillations in Alg. 2, line 1 (see Fig. 9 in the appendix for more information).

In view of the definitions for $c, c_i$, the local gradient (4) can be re-written as:

$$\nabla G_i(w_i^{t,e}; w_{-i}^t) = \nabla f_i(w_i^{t,e}) + (1 - \gamma_i)(c^t - c_i^{t,e}) + (1 - \gamma_i)\frac{1}{m}(c_i^{t,e} - c_i^t) + \rho w_i^{t,e}. \quad (7)$$

The derivation is provided in Appendix A. Consequently, at the beginning of the $t$-th global round of `pFedGT`, the server selects a subset of users (referred to as $S^t$), which subsequently download $c^t$ and $\theta^t$ from the server. Local training based on problem (2) is subsequently performed to update the personalized model using SGD steps as in (7) (Alg. 2, lines 2-8). Following the local training phase, active users upload the difference between successive combined messages as well as the differences in their personalized models which are aggregated by the server (Alg. 1, lines 7-8). The full description of `pFedGT` is shown in Alg. 1 and Alg. 2, and the architecture is shown in Fig. 2.

## 4 ANALYSIS

In this section, we present the convergence analysis for our proposed method. Our analysis requires the following two standard assumptions (the first is about the loss function and the second about the accuracy of solving the local problems).

**Assumption 1.** Each local loss function $f_i(\cdot)$ has an $L$-Lipschitz continuous gradient, i.e., $\forall\, w_i, w_i' \in \mathbb{R}^d$, the following inequality holds:

$$\|\nabla f_i(w_i) - \nabla f_i(w_i')\| \leq L\|w_i - w_i'\|, \ \forall i \in [m].$$

In practical scenarios, problem (2) is not solved exactly, but is rather approximated by running several epochs based on local data. This fact, together with allowing variability of local work across the users (i.e., in view of system heterogeneity) is captured by the next assumption.

**Assumption 2.** At round $t$, each participating user solves problem (2) so that:

$$\left\|\nabla_{w_i} G_i(w_i^{t+1}; w_{-i}^t)\right\|^2 \leq \varepsilon_i.$$

Assumption 1 is a standard and widely used assumption in optimization, and this is the only assumption we need for our analysis. Assumption 2 is non-restrictive because we can establish strong-convexity of (2) in view of Assumption 1, for suitable parameter choices (see Lemma 2 in Appendix B). This, in turn, implies a certain monotonicity of the local gradient for (2) with the local effort. Assumption 2 is only needed to capture the amount of local work: a smaller $\varepsilon_i$ means larger training effort (this is because we can establish strong convexity for the local problems – as a consequence of Lemma 2 in Appendix B – which means that increasing the local work can decrease the gradient $\nabla_{w_i} G_i(w_i; w_{-i})$).

**Theorem 1.** Under Assumption 1, for any $\gamma_i \in (0,1)$, and $\rho > \max\{L, \max_i L_{F_i}\}$, where $L_{F_i} = \frac{(m-1)\gamma_i+1}{m}L + \frac{(1-\gamma_i)(m-1)}{m}\mu$, there exists a unique *Nash equilibrium*, denoted by $w^\star$.

**Theorem 2.** Let Assumptions 1 and 2 hold. For any $\gamma_i \in (0,1)$, $\lambda = |S^t|/m, \rho > \max\{L, \max_i L_{F_i}\}$, and assume each client has a probability of being selected at each round that is lower bounded by a positive constant $p_{\min} > 0$, then the following holds:

$$\mathbb{E}\big[\|w^t - w^\star\|^2\big] \leq \frac{a^t}{p_{\min}}\|w^0 - w^\star\|^2 + \frac{1-a^t}{1-a}\sum_{i=1}^{m}\frac{(1+\xi)\varepsilon_i}{(\rho - L_{F_i})^2},$$

where $a = 1 - p_{min}(1 - (1+\xi^{-1})a_1^2) \in (0,1)$, $\xi$ is an any constant that satisfies $\xi \geq a_1/(1-a_1)$, and $a_1 = \max_i \frac{(1-\gamma_i)(m-1)(\mu+L)}{(1-\gamma_i)(m-1)\mu - (\gamma_i(m-1)+1)L + m\rho}$, $L_{F_i} = \frac{(m-1)\gamma_i+1}{m}L + \frac{(1-\gamma_i)(m-1)}{m}\mu$.

The detailed proofs are provided in Appendix B.

In addition to the Assumptions 1-2, the following assumption has been imposed when analyzing existing state-of-the-art methods to capture the level of statistical heterogeneity. Our analysis *does not require this condition*.

(*Bounded diversity*) The variance of local gradients to global gradient is bounded, i.e., $\frac{1}{m}\sum_{i=1}^{m}\|\nabla f_i(w) - \nabla f(w)\|^2 \leq \sigma^2$, where $f(w) := \frac{1}{m}\sum_{i=1}^{m}f_i(w)$.

**Remark 1.** Our analysis requires no assumption on the level of statistical heterogeneity (i.e., *bounded diversity*, in contrast to several methods in the FL literature (T Dinh et al., 2020; Fallah et al., 2020; Li et al., 2021). Theorem 1 and 2 are established under any selection of $\gamma_i \in (0,1)$, our problem formulation guarantees the existence uniqueness of Nash equilibrium. Further, our algorithm converges linearly to the equilibrium. Moreover, the convergence rate $a = \mathcal{O}(1)$ as a function of the system size $m$, which supports the scalability of the proposed algorithm. Theorem 2 is established with a simple activation scheme of each user is active with probability lower bound by $p_{\min} > 0$. In practice, this assumption is necessary otherwise the user will never participate.

## 5 EXPERIMENTS

All our experiments are conducted on a system with 2 Intel® Xeon® Gold 6330 CPUs and 8 NVIDIA® GeForce RTX™ 3090 GPUs. We compare `pFedGT` against a variety of PFL methods, including `PerFedAvg` (Fallah et al., 2020), `Ditto` (Li et al., 2021), `pFedMe` (T Dinh et al., 2020), `APFL` (Deng et al., 2020), `FedRep` (Collins et al., 2021), `FedBABU` (Oh et al., 2022). For a comprehensive evaluation, we also perform experiments by testing the global model on local test data

Table 1: Average test accuracies for various degrees of non-IID data distributions on CIFAR-10, CIFAR-100 and HAR datasets with participation rate 0.25. $\alpha$ represents the degree of Dirichlet distribution, while $N$ represents the degree of the pathological distribution. The smaller the values of both $\alpha$ and $N$, the greater the heterogeneity in the distribution. All experiments are conducted in the same setting (local epoch number is 5 and batch size is 128). `Local only` refers to a scenario where training is performed solely on local data without any communication or collaboration with the server. The underlined numbers represent the best accuracy across the baseline methods, and the improvement is computed over the best baseline.

| (degree of non-IID) | CIFAR-10 | | | CIFAR-100 | | | HAR |
| | $\alpha = 0.1$ | $\alpha = 1$ | $\alpha = 10$ | $N = 20$ | $N = 50$ | $N = 100$ | – |
|---|---|---|---|---|---|---|---|
| Local only | 87.79% | 61.19% | 48.15% | 53.65% | 38.67% | 29.76% | 62.85% |
| FedAvg | 82.16% | 67.28% | 75.01% | 55.46% | 43.69% | 48.64% | 88.74% |
| FedAvg+FT | 91.26% | 78.34% | 75.39% | 64.48% | 51.89% | 48.73% | 89.59% |
| PerFedAvg | 80.51% | 57.82% | 49.71% | 28.61% | 25.99% | 18.73% | 25.70% |
| Ditto | 88.78% | 73.48% | 75.23% | 57.69% | 48.53% | 49.24% | 86.61% |
| pFedMe | 88.32% | 69.54% | 62.69% | 52.12% | 33.29% | 28.08% | 39.03% |
| APFL | 91.72% | 79.04% | 76.53% | 63.80% | 52.38% | 47.78% | 92.34% |
| FedRep | 91.56% | 80.36% | 76.30% | 65.93% | 54.52% | 44.93% | 78.87% |
| FedBABU | 91.45% | 78.61% | 74.34% | 66.24% | 58.09% | 49.19% | 42.49% |
| pFedGT | **92.64%** | **83.55%** | **80.22%** | **70.57%** | **60.50%** | **51.52%** | **95.51%** |
| **Improvement** | 0.92% | 3.19% | 3.69% | 4.33% | 2.41% | 2.28% | 3.17% |

using both `FedAvg` (McMahan et al., 2017) and its fine-tuning (FT) method (Wang et al., 2019).[1] In brief, our experiments unravel three main findings: (i) `pFedGT` achieves higher average test accuracies across various levels of non-IID data distributions; (ii) `pFedGT` consistently outperforms both local training and conventional FL, thus corroborating the urge for personalization on accounts for statistical heterogeneity; (iii) `pFedGT` users can enhance the local accuracy by increasing the local workload, without hurting the convergence of the algorithm.

## 5.1 EXPERIMENTAL SETUP

Three real datasets are used in our experiments, namely CIFAR-10, CIFAR-100 (Krizhevsky, 2009), and Human Activity Recognition (HAR)[2]. We use ResNet-18 for both CIFAR-10 and CIFAR-100, and a CNN model with two convolutional layers for HAR. In all cases, we set the total number of users equal to 20 ($m = 20$) for CIFAR datasets and 30 ($m = 30$) for HAR dataset. We adopt the random initialization for the personalized models and $c^0 = \sum_{i=1}^{m} c_i^0$. At each round, we fix the selection ratio equal to 0.25, that is, 5 users are selected uniformly at random in CIFAR experiments and 7 in the HAR experiments. We calculate the average of all the users' test accuracy as the comparison metric.

**Hyperparameters:** Drawing inspiration from the experimental results presented in Fig. 1, we conducted a grid search to determine the optimal value for the hyperparameter $\gamma_i$, ranging from 0.1 to 0.9. Subsequently, we selected a fixed value of $\gamma_i = 0.8, \forall i$ (empirically fixed) for all experiments conducted. We tuned the other hyperparameters used in `pFedGT` to choose the empirically best one, e.g., we set $\eta_s = 1$, $\lambda = 0.7$, $\mu = 0.05$, and $\rho = 0$ (we give more explanations for hyperparameter selection in Appendix D). We stop all the algorithms when the number of global rounds reaches 100. By default, we let the local training epoch be fixed to 5, and the batch size equal to 128. We first tune the learning rate of the local SGD solver for `FedAvg` from a candidate set $\{0.01, 0.05, 0.1, 0.5\}$ for best performance, then keep it fixed for all algorithms.

**Data Distribution:** We explore different levels of statistical heterogeneity across the system in two different ways. Note that both of these further yield unequal data volumes across the users. For CIFAR-10 and CIFAR-100 datasets, we adopt the following data partition strategies.

---

[1]We give the detailed description of baseline methods in Appendix C.
[2]https://archive.ics.uci.edu/ml/datasets/

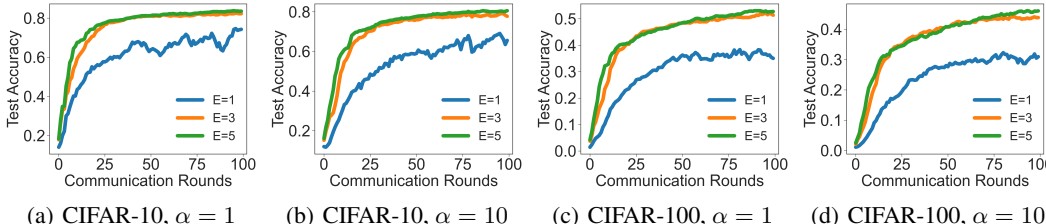

Figure 3: By increasing the number of local epochs $E$, our proposed method `pFedGT` achieves higher accuracy and faster convergence. Moreover, `pFedGT` maintains stability and robustness in all test scenarios.

**1. Pathological distribution:** We sort all the training data by labels and divide them evenly into $N \times m$ shards, and then assign to each user $N$ of these shards. We obtain different levels of non-IID by varying the value of $N$, i.e., a smaller $N$ leads to a higher degree of statistical heterogeneity.

**2. Dirichlet distribution:** We use the Dirichlet distribution as in (Hsu et al., 2019) to create disjoint non-IID client data. We let each user draw training examples independently, and the class labels follow a distribution which is parameterized by a vector $q$ where $q_i \geq 0$ for $i \in [1, N]$ and $\|q\|_1 = 1$. We sample $q$ from a Dirichlet distribution $q \sim Dir(\alpha p)$, where $p$ characterizes a prior class distribution over the $N$ classes, and $\alpha > 0$ is a parameter controlling the degree of similarity between users. A smaller $\alpha$ leads to a higher degree of statistical heterogeneity. To provide a visual representation of the effects of varying $\alpha$, Fig. 7 in Appendix C presents how the samples are distributed among 20 users for different values of $\alpha$ on the CIFAR-10 dataset.

Regarding the HAR dataset, we implement a partitioning strategy in which all data instances produced by a single individual are assigned to a distinct user. This strategy intentionally generates an inherently diverse data distribution. Our code is available at: `anonymous.4open.science/r/pFedGT`.

## 5.2 EXPERIMENTAL RESULTS

We first demonstrate the algorithm performance using the averaged test accuracy after 100 global rounds. The results are summarized in Table 1. We note that `pFedGT` consistently outperforms all baseline methods in all cases tested. Specifically, it achieves on average 2.6% and 3.0% higher accuracy compared to the second based baseline in CIFAR-10 and in CIFAR-100, respectively, and 3.17% higher in the HAR dataset.

**Increasing Local Work.** We investigate the effect of local computation on the performance of `pFedGT` by increasing the local epoch number $E$. As shown in Fig. 3, we observe that increasing the amount of local training leads to faster convergence and higher accuracy, indicating the effectiveness of the personalized approach. Additionally, we compare `pFedGT` with `FedAvg+FT` when varying the local epoch number: Fig. 4 demonstrates that the performance of `FedAvg+FT` may decrease with an increased workload, while `pFedGT` consistently maintains good performance by increasing the local workload.

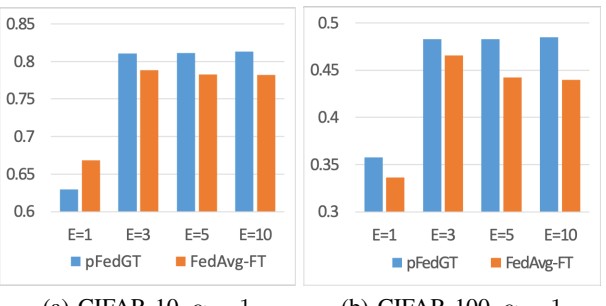

Figure 4: Test accuracy for variable number of local epochs. `pFedGT` showcased a consistent improvement for increased $E$, while this is not the case for `FedAvg+FT`.

**Local Initialization.** We conducted a series of experiments to investigate the impact of different local initialization approaches on the performance of `pFedGT`. Given that the local sub-optimization problem (2) is an unconstrained problem (which is further shown to be strongly convex—see Appendix B), users have the flexibility to choose various starting points for their local training. Specifically, we examined three initialization strategies: (i) $\theta$ initialization, as proposed in our algorithm, (ii) $w_i$ initialization, corresponding to initializing with the personalized model $w_i$, and (iii) random

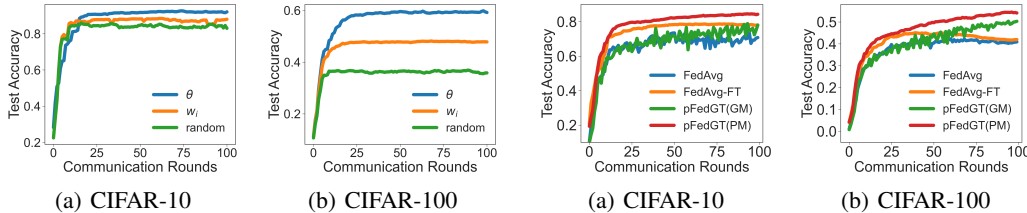

(a) CIFAR-10     (b) CIFAR-100     (a) CIFAR-10     (b) CIFAR-100

Figure 5: Different approaches on local initialization Figure 6: Experiments are conducted in a level of $\alpha = 1$ for personalized training in the $\alpha = 0.1$ level of non- non-IID. PM stands for accuracy tested using person-IID. Among these approaches, $\theta$ initialization provides alized models, while GM stands for the server model the best warm start for users and yields the best overall $\theta$. Results demonstrate that our server model achieves performance. better results than the global model trained through `FedAvg`.

initialization. The experimental results, depicted in Fig. 5, are in support of our choice to maintain a global model (albeit not necessary for the operation of the algorithm, which we discuss next).

**The Server Model $\theta$.** Holding a global model is not mandatory for `pFedGT` and we can obtain two variants: one with minimal communication cost (no global model, orange line in Fig. 5) and one with superior accuracy at the cost of communicating model parameters and maintaining a global model (Alg. 1, blue line in Fig. 5). In addition to the experiments conducted on initialization, we have also investigated the utility of the server model in `pFedGT` for local tasks. Specifically, we have evaluated the server model $\theta$ on the local dataset and compared it with the standard `FedAvg` algorithm. The results are presented in Fig. 6, where (GM) denotes the global (server) model in `pFedGT` and (PM) represents the personalized models. The notable observation is that `pFedGT(GM)` can obtain superior accuracy compared to the conventional `FedAvg`. Combined with the results of the local initialization experiments, it suggests that a new user can rapidly catch up with the federated system to achieve personalization.

**Scalability.** To exhibit the robustness and scalability of `pFedGT`, we performed experiments for both CIFAR-10 and CIFAR-100 datasets in a larger system comprising of 100 users. Table 2 presents the experimental results in different pathological data distribution scenarios. In each communication round, 25 users are selected uniformly at random. Empirical results reveal that the proposed `pFedGT` consistently outperforms the existing state-of-the-art methods in terms of averaged accuracy, particularly in the more challenging dataset (CIFAR-100). This performance differential validates the efficacy and robustness of `pFedGT` in large-scale FL systems, underscoring its potential as a formidable tool in advancing the performance and scalability of FL applications.

Table 2: Experiments conducted on a system of 100 users. The results consistently demonstrate that `pFedGT` outperforms other PFL methods in terms of accuracy.

| | CIFAR-10 | | CIFAR-100 | |
|---|---|---|---|---|
| (degree of non-IID) | $N = 2$ | $N = 5$ | $N = 5$ | $N = 20$ |
| `PerFedAvg` | 84.01% | 82.83% | 78.60% | 61.74% |
| `Ditto` | 87.44% | 82.88% | 79.48% | 56.34% |
| `pFedMe` | 77.76% | 68.08% | 70.32% | 40.37% |
| `APFL` | 90.05% | 84.56% | 79.72% | 58.66% |
| `FedRep` | 90.30% | 85.50% | 80.34% | 60.35% |
| `FedBABU` | 89.88% | 85.40% | 81.93% | 63.22% |
| `pFedGT` | **90.55%** | **85.58%** | **82.97%** | **66.22%** |

## 6 CONCLUSION

In this paper, we proposed `pFedGT`, a novel game-theoretic approach for PFL that directly interpolates the local and global targets in the domain field which we coined "target interpolation". An efficient federated implementation is obtained by second-order approximation of the global term, thus a game is formulated where the user (selfishly) minimizes their reformulated loss without considering the impact on others. We prove existence and uniqueness of a *Nash equilibrium* and establish a linear convergence rate in a general nonconvex setting under standard assumptions. Extensive comparative experiments were used to corroborate the merits of `pFedGT` as a well-suited candidate solution for PFL.

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

APPENDIX This appendix is composed of four parts: (i) the full derivation of equation (7) in Section 3.2; (ii) the full proof of the theoretical results presented in Section 4; (iii) detailed experimental settings that were omitted due to the space limitations in the main paper; (iv) additional experiments that demonstrate various aspects of the proposed algorithm.

## A DERIVATION OF LOCAL GRADIENTS (EQUATION (7))

In this section, we give the detailed derivation of equation (7). First, we elaborate on the settings for the number of global rounds $t$ and local epochs $e \in [0, E-1]$. At global round $t$, local epoch $e$ for user $i$, we have:

$$\nabla_{w_i} G_i(w_i^{t,e}; w_{-i}^t) = \gamma_i \nabla f_i(w_i^{t,e}) + \frac{1-\gamma_i}{m} \nabla f_i(w_i^{t,e}) + \frac{1-\gamma_i}{m} \sum_{j \neq i}^{m} \left( \nabla f_j(w_j^t) + \mu(w_i^{t,e} - w_j^t) \right) + \rho w_i^{t,e}$$

$$= \gamma_i \nabla f_i(w_i^{t,e}) + \frac{1-\gamma_i}{m} \nabla f_i(w_i^{t,e}) + \frac{(1-\gamma_i)(m-1)}{m} \mu w_i^{t,e}$$

$$+ \underbrace{\frac{1-\gamma_i}{m} \sum_{j \neq i}^{m} \left( \nabla f_i(w_j^t) - \mu w_j^t \right)}_{(i)} + \rho w_i^{t,e}.$$

By adding and subtracting $\nabla f_i(w_i^t) - \mu w_i^t$ in (i) we obtain:

$$\nabla_{w_i} G_i(w_i^{t,e}; w_{-i}^t) = \gamma_i \nabla f_i(w_i^{t,e}) + \frac{1-\gamma_i}{m} \nabla f_i(w_i^{t,e}) + \frac{(1-\gamma_i)(m-1)}{m} \mu w_i^{t,e}$$

$$+ (1-\gamma_i) \frac{1}{m} \sum_{j=1}^{m} \left( \nabla f_i(w_j^t) - \mu w_j^t \right) - \frac{1-\gamma_i}{m} \left( \nabla f_i(w_i^t) - \mu w_i^t \right) + \rho w_i^{t,e}$$

$$= \gamma_i \nabla f_i(w_i^{t,e}) + (1-\gamma_i) \mu w_i^{t,e} + \frac{1-\gamma_i}{m} \nabla f_i(w_i^{t,e}) - \frac{1-\gamma_i}{m} \mu w_i^{t,e}$$

$$+ (1-\gamma_i) \frac{1}{m} \sum_{j=1}^{m} \left( \nabla f_i(w_j^t) - \mu w_j^t \right) - \frac{1-\gamma_i}{m} \left( \nabla f_i(w_i^t) - \mu w_i^t \right) + \rho w_i^{t,e}$$

$$= \nabla f_i(w_i^{t,e}) - (1-\gamma_i) \left( \nabla f_i(w_i^{t,e}) - \mu w_i^{t,e} \right) + \frac{1-\gamma_i}{m} \left( \nabla f_i(w_i^{t,e}) - \mu w_i^{t,e} \right) + \rho w_i^{t,e}$$

$$+ (1-\gamma_i) \frac{1}{m} \sum_{j=1}^{m} \left( \nabla f_i(w_j^t) - \mu w_j^t \right) - \frac{1-\gamma_i}{m} \left( \nabla f_i(w_i^t) - \mu w_i^t \right) + \rho w_i^{t,e}$$

$$\underset{(ii)}{=} \nabla f_i(w_i^{t,e}) + (1-\gamma_i)(c^t - c_i^{t,e}) + (1-\gamma_i) \frac{1}{m}(c_i^{t,e} - c_i^t) + \rho w_i^{t,e},$$

where we use the definitions $c_i := \nabla f_i(w_i) - \mu w_i$ and $c := \frac{1}{m} \sum_{i=1}^{m} c_i$ in (ii).

## B COMPLETE PROOF OF THE RESULTS

Before presenting the proofs of theorems 1 and 2, we introduce the following standard propositions, which are frequently used in our analysis.

**Proposition 1.** If a function $f(x)$ has an $L$-Lipschitz continuous gradient, then $g(x) := f(x) + \frac{k}{2}\|x\|^2$ is $(k-L)$-strongly convex for any constant $k > L$.

**Proposition 2.** If a function $f(x)$ is $\sigma$-strongly convex, then the operator $T := (I_d + \lambda \nabla f)^{-1}$ is contractive with coefficient $\frac{1}{1+\lambda\sigma}$, $\forall \lambda > 0$.

**Proposition 3.** If a function $f(x)$ has an $L$-Lipschitz continuous gradient and is $\sigma$-strongly convex, then the following inequalities hold:

$$\langle \nabla f(x) - \nabla f(y), x - y \rangle \geq \frac{\sigma L}{\sigma + L} \|x - y\|^2 + \frac{1}{\sigma + L} \|\nabla f(x) - \nabla f(y)\|^2 \tag{8a}$$

$$\|\nabla f(x) - \nabla f(y)\| \geq \sigma \|x - y\| \tag{8b}$$

**Lemma 1.** Let $\lambda = |S^t|/m$, then under (5), by initilizing $c^0 = \sum_{i=0}^{m} c_i^0$, it holds that

$$c^t = \sum_{i=1}^{m} c_i^t, \forall t.$$

*Proof:* Since $\lambda = |S^t|/m$, we have that $c^{t+1} = c^t + \frac{1}{m} \sum_{i \in S^t} (c_i^{t+1} - c_i^t)$. Due to the partial participation in FL, the following holds

$$c_i^{t+1} = \begin{cases} \nabla f_i(w_i^{t+1}) - \mu w_i^{t+1}, & \text{if } i \in S^t, \\ c_i^t, & \text{if } i \notin S^t. \end{cases} \tag{9}$$

Equation (9) shows that if user $i$ is active, it continues to update its communication messages; otherwise, $c_i$ remains unchanged, retaining the value from the end of the previous active global round. We obtain $c^{t+1} = c^t + \frac{1}{m} \sum_{i=1}^{m} (c_i^{t+1} - c_i^t)$. After telescoping, we obtain $c^t = c^0 + \frac{1}{m} \sum_{i=1}^{m} (c_i^t - c_i^0)$. Together with the initialization, $c^0 = \sum_{i=1}^{m} c_i^0$, the desired is proved. ∎

Lemma 1 indicates that in every global round, under any selection strategy mechanisms, the variable $c$ consistently represents the precise average of all users' $c_i$. Supporting by lemma 1, we can establish the following theorems. For simplicity, we denote $w$ as the concatenation of all the personalized models $w_i$, and $w \in \mathbb{R}^{md}$.

**Lemma 2.** Let assumption 1 hold, then each $F_i(w_i; w_{-i})$ has a $L_{F_i}$-Lipschitz continuous gradient in terms of $w_i$, where $L_{F_i} = \frac{(m-1)\gamma_i + 1}{m} L + \frac{(1-\gamma_i)(m-1)}{m} \mu$. In particular, $\rho > L_{F_i}$, $G_i(w_i; w_{-i})$ is $(\rho - L_{F_i})$-strongly convex.

*Proof:* From the definition of $F_i(w_i; w_{-i})$:

$$\nabla_{w_i} F_i(w_i; w_{-i}) = \gamma_i \nabla f_i(w_i) + \frac{1 - \gamma_i}{m} \sum_{j \neq i} \left( \nabla f_j(w_j) + \mu(w_i - w_j) \right) + \frac{1 - \gamma_i}{m} \nabla f_i(w_i)$$

$$= \frac{(m-1)\gamma_i + 1}{m} \nabla f_i(w_i) + \frac{(1-\gamma_i)(m-1)}{m} \mu w_i + \frac{1 - \gamma_i}{m} \sum_{j \neq i} \left( \nabla f_j(w_j) - \mu w_j \right).$$

then $\forall w_i, w_i' \in \mathbb{R}^d$, we obtain:

$$\left\| \nabla_{w_i} F_i(w_i; w_{-i}) - \nabla_{w'} F_i(w_i'; w_{-i}) \right\| = \left\| \left( \frac{(m-1)\gamma_i + 1}{m} \nabla f_i(w_i) + \frac{(1-\gamma_i)(m-1)}{m} \mu w_i \right) \right.$$

$$\left. - \left( \frac{(m-1)\gamma_i + 1}{m} \nabla f_i(w_i') + \frac{(1-\gamma_i)(m-1)}{m} \mu w_i' \right) \right\|$$

$$\leq \frac{(m-1)\gamma_i + 1}{m} \left\| \nabla f_i(w_i) - \nabla f_i(w_i') \right\| + \frac{(1-\gamma_i)(m-1)}{m} \mu \left\| w_i - w_i' \right\|$$

$$\leq \left( \frac{(m-1)\gamma_i + 1}{m} L + \frac{(1-\gamma_i)(m-1)}{m} \mu \right) \left\| w_i - w_i' \right\|.$$

We denote $L_{F_i} = \frac{(m-1)\gamma_i + 1}{m} L + \frac{(1-\gamma_i)(m-1)}{m} \mu$, and the second claim holds by Proposition 1. ∎

### B.1 PROOF OF THEOREM 1

**Theorem 1.** Under Assumption 1, for $\rho > \max\{L, \max_i L_{F_i}\}$, where $L_{F_i} = \frac{(m-1)\gamma_i + 1}{m} L + \frac{(1-\gamma_i)(m-1)}{m} \mu$, there exists a unique *Nash equilibrium*, denoted by $w^\star$.

*Proof:* Following Lemma 2, for $\rho > \max_i L_{F_i}$, each $G_i(w_i; w_{-i})$ is $(\rho - L_{F_i})$-strongly convex with repect to $w_i$. We define the operator $R_i(w) : \mathbb{R}^{md} \rightarrow \mathbb{R}^d$ as:

$$R_i(w) = \arg \min_{w_i} G_i(w_i; w_{-i}),$$

i.e., the set of minimizers of (2). In view of Lemma 2, this is single-valued, i.e., a function (the strong convexity implies uniqueness of solution), we further define $R(w) : \mathbb{R}^{md} \rightarrow \mathbb{R}^{md}$ to be the concatenation of all $R_i(w)$. The proof will be carried by establishing that $R$ is contractive.

The optimality condition for (2) is given by:

$$
\begin{aligned}
\nabla_{w_i} G_i(w_i; w_{-i}) = {} & \gamma_i \nabla f_i(w_i) + \frac{1 - \gamma_i}{m} \nabla f_i(w_i) + \frac{(1 - \gamma_i)(m - 1)}{m} \mu w_i + \rho w_i \\
& + \frac{1 - \gamma_i}{m} \sum_{j \neq i}^{m} \left( \nabla f_i(w_j) - \mu w_j \right) \\
= {} & \frac{(m - 1)\gamma_i + 1}{m} \nabla f_i(w_i) + \frac{(1 - \gamma_i)(m - 1)}{m} \mu w_i + \rho w_i \\
& + \frac{1 - \gamma_i}{m} \sum_{j \neq i} \left( \nabla f_j(w_j) - \mu w_j \right) = 0,
\end{aligned}
$$

For ease of exposition, we define $\tilde{\gamma}_i = \frac{(m-1)\gamma_i + 1}{m}, \tilde{\gamma}_i \in (\frac{1}{m}, 1)$, then $\frac{(1-\gamma_i)(m-1)}{m} = 1 - \tilde{\gamma}_i$, and denote $g_i(x_i) := f_i(x_i) + \frac{\rho}{2\tilde{\gamma}_i} \|x_i\|^2$. Since we select $\rho > L \Rightarrow \frac{\rho}{\tilde{\gamma}_i} > L$. By Proposition 1, $g_i(x)$ is $(\frac{\rho}{\tilde{\gamma}_i} - L)$-strongly convex. Then we obtain:

$$
\tilde{\gamma}_i \nabla f_i(w_i) + (1 - \tilde{\gamma}_i) \mu w_i + \rho w_i + \frac{1 - \tilde{\gamma}_i}{m - 1} \sum_{j \neq i}^{m} (\nabla f_j(w_j) - \mu w_j) = 0
$$

$$
\Rightarrow \quad (1 - \tilde{\gamma}_i)\mu \left( I_d + \frac{\tilde{\gamma}_i}{(1 - \tilde{\gamma}_i)\mu} \nabla g_i \right)(w_i) = \frac{1 - \tilde{\gamma}_i}{m - 1} \sum_{j \neq i}^{m} (\mu I_d - \nabla f_j)(w_j),
$$

Here, $I_d$ represents the identity operator ($I_d x = x$). The resolvent operator is defined as:

$$
T_i := \left( I_d + \frac{\tilde{\gamma}_i}{(1 - \tilde{\gamma}_i)\mu} \nabla g_i \right)^{-1},
$$

which is contractive with coefficient $\rho_1 := \frac{1}{1 + \frac{\tilde{\gamma}_i}{(1 - \tilde{\gamma}_i)\mu}(\frac{\rho}{\tilde{\gamma}_i} - L)} < 1$ by Proposition 2.

This implies that

$$
R_i(w) = T_i \cdot \frac{1}{\mu(m - 1)} \sum_{j \neq i}^{m} (\mu I_d - \nabla f_j)(w_j),
$$

where $w$ is the concatenation of $w_i$. Then $\forall w, w' \in \mathbb{R}^{md}$, we obtain:

$$
\begin{aligned}
\|R_i(w) - R_i(w')\|^2 &= \left\| T_i \cdot \frac{1}{\mu(m - 1)} \sum_{j \neq i}^{m} (\mu I_d - \nabla f_j)(w_j) - T_i \cdot \frac{1}{\mu(m - 1)} \sum_{j \neq i}^{m} (\mu I_d - \nabla f_j)(w_j') \right\|^2 \\
&\underset{\text{(i)}}{\leq} \rho_1^2 \left\| \frac{1}{\mu(m - 1)} \sum_{j \neq i}^{m} (\mu I_d - \nabla f_j)(w_j) - \frac{1}{\mu(m - 1)} \sum_{j \neq i}^{m} (\mu I_d - \nabla f_j)(w_j') \right\|^2 \\
&= \frac{\rho_1^2}{\mu^2(m - 1)^2} \left\| \sum_{j \neq i}^{m} \left( (\mu I_d - \nabla f_j)(w_j) - (\mu I_d - \nabla f_j)(w_j') \right) \right\|^2. \qquad (10)
\end{aligned}
$$

The inequality (i) follows from the fact that $T_i$ is a contraction with coefficient $\rho_1 = \frac{1}{1 + \frac{\tilde{\gamma}_i}{(1 - \tilde{\gamma}_i)\mu}(\frac{\rho}{\tilde{\gamma}_i} - L)}$.

We further define the operator $P_j(w_j) := (\mu I_d - \nabla f_j)(w_j)$ and $h_j(w_j) := f_j(w_j) + \frac{k}{2}\|w_j\|^2, \forall j$, where $k$ is any constant that satisfies $k > L$. By Proposition 1, $h_j(w_j)$ is $(k - L)$-strongly convex, and its gradient is Lipschitz continuous with parameter $k + L$. We adopt the following equivalent expression for $P_j$:

$$
\begin{aligned}
P_j &= \mu I_d - \nabla f_j = (\mu + k) I_d - (\nabla f_j + k I_d) \\
&= (\mu + k) I_d - \nabla h_j \\
&= (\mu + k)(I_d - \frac{\nabla h_j}{\mu + k}).
\end{aligned}
$$

Consequently, it holds that:

$$
\begin{aligned}
||P_j(w_j) - P_j(w_j')||^2 &= (\mu+k)^2||(I_d - \frac{\nabla h_j}{\mu+k})(w_j) - (I_d - \frac{\nabla h_j}{\mu+k})(w_j')||^2 \\
&= (\mu+k)^2||(w_j - w_j') - \frac{1}{\mu+k}(\nabla h_j(w_j) - \nabla h_j(w_j'))||^2 \\
&= (\mu+k)^2\Big(||w_j - w_j'||^2 + \frac{1}{(\mu+k)^2}||\nabla h_j(w_j) - \nabla h_j(w_j')||^2 \\
&\quad - \frac{2}{\mu+k}\langle \nabla h_j(w_j) - \nabla h_j(w_j'), w_j - w_j'\rangle\Big) \\
&\underset{(i)}{\leq} (\mu+k)^2\Big(||w_j - w_j'||^2 + \frac{1}{(\mu+k)^2}||\nabla h_j(w_j) - \nabla h_j(w_j')||^2 \\
&\quad - \frac{k^2 - L^2}{(\mu+k)k}||w_j - w_j'||^2 - \frac{1}{(\mu+k)k}||\nabla h_j(w_j) - \nabla h_j(w_j')||^2\Big) \\
&\leq (\mu+k)^2\Big((1 - \frac{k^2 - L^2}{(\mu+k)k})||w_j - w_j'||^2 - \frac{\mu}{(\mu+k)^2 k}||\nabla h_j(w_j) - \nabla h_j(w_j')||^2\Big) \\
&\underset{(ii)}{\leq} (\mu+k)^2\Big((1 - \frac{k^2 - L^2}{(\mu+k)k} - \frac{\mu(k-L)^2}{(\mu+k)^2 k})||w_j - w_j'||^2\Big) \\
&= (\mu+L)^2||w_j - w_j'||^2. \tag{11}
\end{aligned}
$$

Inequalities (i) and (ii) follow from Proposition 3, i.e., (8a) and (8b), respectively.

In turn, it follows that:

$$
\begin{aligned}
||R_i(w) - R_i(w')||^2 &\leq \frac{\rho_1^2}{\mu^2(m-1)^2}\Big\|\sum_{j\neq i}^{m}\Big((\mu I_d - \nabla f_j)(w_j) - (\mu I_d - \nabla f_j)(w_j')\Big)\Big\|^2. \\
&\underset{(i)}{\leq} \frac{\rho_1^2}{\mu^2(m-1)}\sum_{j\neq i}^{m}||P_j(w_j) - P_j(w_j')||^2 \\
&\underset{(ii)}{\leq} \frac{\rho_1^2}{\mu^2}\cdot(\mu+L)^2\frac{1}{m-1}\sum_{j\neq i}^{m}||w_j - w_j'||^2 \\
&\underset{(iii)}{=} \frac{(1-\tilde{\gamma}_i)^2\mu^2}{\big((1-\tilde{\gamma}_i)\mu + \tilde{\gamma}_i(\frac{\rho}{\tilde{\gamma}_i} - L)\big)^2}\frac{(\mu+L)^2}{\mu^2}\frac{1}{m-1}\sum_{j\neq i}^{m}||w_j - w_j'||^2 \\
&= \frac{(1-\tilde{\gamma}_i)^2(\mu+L)^2}{\big((1-\tilde{\gamma}_i)\mu + \tilde{\gamma}_i(\frac{\rho}{\tilde{\gamma}_i} - L)\big)^2}\frac{1}{m-1}\sum_{j\neq i}^{m}||w_j - w_j'||^2,
\end{aligned}
$$

where (i) is by an application of Jensen's inequality $\|\sum_{i=1}^{n} w_i\|^2 \leq n\sum_{i=1}^{n}\|w_i\|^2$, and (ii) follows from (11); Substituting $\rho_1 = \frac{1}{1+\frac{\tilde{\gamma}_i}{(1-\tilde{\gamma}_i)\mu}(\frac{\rho}{\tilde{\gamma}_i}-L)}$, we get (iii).

By selecting $\rho > L$, we obtain:

$$
\frac{(1-\tilde{\gamma}_i)^2(\mu+L)^2}{\big((1-\tilde{\gamma}_i)\mu + \tilde{\gamma}_i(\frac{\rho}{\tilde{\gamma}_i} - L)\big)^2} < \frac{(1-\tilde{\gamma}_i)^2(\mu+L)^2}{\big((1-\tilde{\gamma}_i)\mu + \tilde{\gamma}_i(\frac{L}{\tilde{\gamma}_i} - L)\big)^2} = \frac{(1-\tilde{\gamma}_i)^2(\mu+L)^2}{\big((1-\tilde{\gamma}_i)\mu + (1-\tilde{\gamma}_i)L\big)^2} = 1.
$$

We conclude:

$$
\begin{aligned}
||R(w) - R(w')||^2 = \sum_{i=1}^{m}||R_i(w) - R_i(w')||^2 &\leq \sum_{i=1}^{m}\frac{(1-\tilde{\gamma}_i)^2(\mu+L)^2}{\big((1-\tilde{\gamma}_i)\mu + \tilde{\gamma}_i(\frac{\rho}{\tilde{\gamma}_i} - L)\big)^2}\frac{1}{m-1}\sum_{j\neq i}^{m}||w_j - w_j'||^2 \\
&\leq a_1^2\sum_{i=1}^{m}\frac{1}{m-1}\sum_{j\neq i}^{m}||w_j - w_j'||^2 \leq \alpha^2\|w - w'\|^2.
\end{aligned}
$$

where $a_1 = \max_i \frac{(1-\tilde{\gamma}_i)(\mu+L)}{\left((1-\tilde{\gamma}_i)\mu-\tilde{\gamma}_i L+\rho\right)} = \max_i \frac{(1-\gamma_i)(m-1)(\mu+L)}{(1-\gamma_i)(m-1)\mu-(\gamma_i(m-1)+1)L+m\rho} < 1$, by selecting $\rho > \max\{L, \max_i\{\frac{(m-1)\gamma_i+1}{m}L + \frac{(1-\gamma_i)(m-1)}{m}\mu\}\}$.

This is equivalent to

$$\|R(x) - R(y)\| \leq a_1\|x - y\|, \forall x, y, \tag{12}$$

which shows that the operator $R$ is contractive. From Banach fixed-point theorem, it follows that there exists a unique fixed point $w^\star$, which is the unique *Nash equilibrium*. ∎

### B.2  PROOF OF THEOREM 2

Before we present our analysis for our algorithm, we first introduce some notations used to present partial participation. First, we define $\|x\|_A := \sqrt{x^\top A x}$, for any positive definite matrix $A \succ 0$. $\Lambda^t \in \mathbb{R}^{md \times md}$ is a diagonal random matrix, which denotes the activation matrix with subblocks $\Lambda_{ii} \in \mathbb{R}^{d \times d}, i \in m$, taking values $I_d$ or the zero matrix.

**Theorem 2.** Let Assumptions 1 and 2 hold. For any $\gamma_i \in (0,1)$, $\lambda = |S^t|/m, \rho > \max\{L, \max_i L_{F_i}\}$, and assume each client has a probability of being selected at each round that is lower bounded by a positive constant $p_{\min} > 0$, then the following holds:

$$\mathbb{E}\big[\|w^t - w^\star\|^2\big] \leq \frac{a^t}{p_{\min}}\|w^0 - w^\star\|^2 + \frac{1-a^t}{1-a}\sum_{i=1}^m \frac{(1+\xi)\varepsilon_i}{(\rho - L_{F_i})^2},$$

where $a = 1 - p_{min}(1 - (1+\xi^{-1})a_1^2) \in (0,1)$, $\xi$ is an any constant that satisfies $\xi \geq a_1/(1-a_1)$, and $a_1 = \max_i \frac{(1-\gamma_i)(m-1)(\mu+L)}{(1-\gamma_i)(m-1)\mu-(\gamma_i(m-1)+1)L+m\rho}$, $L_{F_i} = \frac{(m-1)\gamma_i+1}{m}L + \frac{(1-\gamma_i)(m-1)}{m}\mu$.

*Proof:* We define the error term at global round $t$ as the following:

$$e^t := w^{t+1} - R(w^t).$$

Then we have

$$\|e^t\|^2 = \|w^{t+1} - R(w^t)\|^2 = \|w^{t+1} - w^{t+1,\star}\|^2 = \sum_{i=1}^m \|w_i^{t+1} - w_i^{t+1,\star}\|^2$$

$$\underset{(i)}{\leq} \sum_{i=1}^m \frac{1}{(\rho - L_{F_i})^2}\|\nabla_{w_i} G_i(w_i^{t+1}; w_i^t)\|^2 \underset{(ii)}{\leq} \sum_{i=1}^m \frac{\varepsilon_i}{(\rho - L_{F_i})^2},$$

where $w^{t+1,\star}$ is the unique solution of (2) at global round $t$. Inequality (i) follows from Proposition 3, inequality (8b) by selecting $\rho > L_{F_i}$ so that $G_i(w_i; w_{-i})$ is $(\rho - L_{F_i})$-strongly convex with respect to $w_i$; (ii) follows from Assumption 2.

First, we denote the partial participation update rule as the following:

$$w^{t+1} = w^t + \Lambda^t(R(w^t) + e^t - w^t),$$

Under any activation scheme such that $\mathbb{E}^t[\Lambda^{t+1}] = \Lambda \succ 0$, then we have

$$\|w^{t+1} - w^\star\|_{\Lambda^{-1}}^2 = \|w^t + \Lambda^t(R(w^t) + e^t - w^t) - w^\star\|_{\Lambda^{-1}}^2$$

$$= \|w^t - w^\star\|_{\Lambda^{-1}}^2 + 2(w^t - w^\star)^\top \Lambda^{-1}\Lambda^t(R(w^t) + e^t - w^t)$$

$$+ (R(w^t) + e^t - w^t)^\top \Lambda^t \Lambda^{-1}\Lambda^t(R(w^t) + e^t - w^t).$$

Since $\Lambda^{-1}, \Lambda^t$ are diagonal matrices, they commute with each other. From the definition of $\Lambda^t$, we can get that $\Lambda^t\Lambda^t = \Lambda^t$. After taking conditional expectations on both sides, we obtain:

$$\mathbb{E}^t\big[\|w^{t+1} - w^\star\|_{\Lambda^{-1}}^2\big] = \|w^t - w^\star\|_{\Lambda^{-1}}^2 + \|R(w^t) + e^t - w^t\|^2 + 2(w^t - w^\star)^\top(R(w^t) + e^t - w^t)$$

$$\underset{(i)}{=} \|w^t - w^\star\|_{\Lambda^{-1}}^2 + \|R(w^t) + e^t - w^\star\|^2 - \|w^t - w^\star\|^2$$

$$\underset{(ii)}{\leq} \|w^t - w^\star\|_{\Lambda^{-1}}^2 - (1 - (1+\xi^{-1})a_1^2)\|w^t - w^\star\|^2 + (1+\xi)\|e^t\|^2 \tag{13}$$

$$\leq (1 - p_{min}(1 - (1+\xi^{-1})a_1^2))\|w^t - w^\star\|_{\Lambda^{-1}}^2 + (1+\xi)\|e^t\|^2$$

$$\leq (1 - p_{min}(1 - (1+\xi^{-1})a_1^2))\|w^t - w^\star\|_{\Lambda^{-1}}^2 + \sum_{i=1}^m \frac{(1+\xi)\varepsilon_i}{(\rho - L_{F_i})^2},$$

Where (i) is directly obtained by completing squares; (ii) follows from the inequality $||x + y||^2 \leq (1+\xi^{-1})||x||^2 + (1+\xi)||y||^2, \forall \xi > 0$ and (12), and $\mathbb{E}^t[\Lambda^{t+1}] = p_i I_d$, here $I_d \in \mathbb{R}^{d \times d}$ is an identity matrix, and $\xi$ is any positive number, $p_{\min} = \min_i p_i$, which is the minimal active probability across all the users. For ease of exposition, we denote $a = (1 - p_{min}(1 - (1+\xi^{-1})a_1^2))$. Finally taking the expectation of all randomness and by induction, we get

$$\mathbb{E}\big[||w^t - w^\star||^2_{\Lambda^{-1}}\big] \leq a\mathbb{E}||w^{t-1} - w^\star||^2_{\Lambda^{-1}} + \sum_{i=1}^{m} \frac{(1+\xi)\varepsilon_i}{(\rho - L_{F_i})^2}$$

$$\leq a\big(a\mathbb{E}||w^{t-2} - w^\star||^2_{\Lambda^{-1}} + \sum_{i=1}^{m} \frac{(1+\xi)\varepsilon_i}{(\rho - L_{F_i})^2}\big) + \sum_{i=1}^{m} \frac{(1+\xi)\varepsilon_i}{(\rho - L_{F_i})^2}$$

$$\leq \ldots \leq a^t \mathbb{E}||w^0 - w^\star||^2_{\Lambda^{-1}} + (a^{t-1} + a^{t-2} + \ldots + 1)\sum_{i=1}^{m} \frac{(1+\xi)\varepsilon_i}{(\rho - L_{F_i})^2}$$

$$= a^t ||w^0 - w^\star||^2_{\Lambda^{-1}} + \frac{1 - a^t}{1 - a}\sum_{i=1}^{m} \frac{(1+\xi)\varepsilon_i}{(\rho - L_{F_i})^2}$$

Here, $\xi$ is an any positive number that satisfy $1 + \xi^{-1} \leq a_1^{-1}$, which is $\xi \geq a_1/(1 - a_1)$, we have $a = 1 - p_{min}(1 - (1+\xi^{-1})a_1^2) \in (0, 1), \forall p_{\min}$.

Thus, Theorem 2 can be directly obtained by $||w^t - w^\star||^2 \leq ||w^t - w^\star||^2_{\Lambda^{-1}} \leq \frac{1}{p_{\min}}||w^t - w^\star||^2$. ∎

The *constant* $\xi$ here is only needed in our proof, it follows from the inequality $\|x + y\|^2 \leq (1 + \xi^{-1})\|x\|^2 + (1 + \xi)\|y\|^2, \forall \xi > 0$ where we used in inequality (13). Our theorems provide the guarantees of the effectiveness of our method under a large enough $\rho$. However, the analysis is conservative, in that it does not exclude the other choices of $\rho$.

**Discussions.** The Lipschitz continuity is the only assumption we need to prove the existence and uniqueness of the solution under suitable selection of the hyperparameter $\rho$ (as in Theorem 1). Furthermore, our algorithm converges linearly to that solution while the other PFL methods demonstrate a sublinear convergence rate (ditto (Li et al., 2021), pFedMe (T Dinh et al., 2020), PerFedAvg (Fallah et al., 2020)). There is no need for assumptions on the level of statistical heterogeneity (i.e., *bounded diversity*, $\frac{1}{m}\sum_{i=1}^{m}\|\nabla f_i(w) - \nabla f(w)\| \leq \sigma^2$, where $f(w) := \frac{1}{m}\sum_{i=1}^{m} f_i(w)$), which is commonly used in many PFL methods like Ditto, pFedMe, PerFedAvg. In contrast to other papers, which first prove their global model converges, and then prove that their personalized models stay close to their global model, our method directly proves all the personalized models converge linearly (to a unique Nash equilibrium that depends only on the selection of $\gamma_i$ by the users). We summarize the comparison in Table 3.

Table 3: Comparison of theoretical results. LC and Heterogeneity represent whether need to assume Lipschitz continuity and the level of statistical heterogeneity, respectively, while CR represents the convergence rate ($a < 1$).

| Algorithm | LC | Heterogeneity | CR |
|---|---|---|---|
| ditto | ✓ | ✓ | $\mathcal{O}(\frac{1}{T})$ |
| pFedMe | ✓ | ✓ | $\mathcal{O}(\frac{1}{T})$ |
| PerFedAvg | ✓ | ✓ | $\mathcal{O}(\frac{1}{T})$ |
| pFedGT | ✓ | - | $\mathcal{O}(a^T), \alpha \in (0, 1)$ |

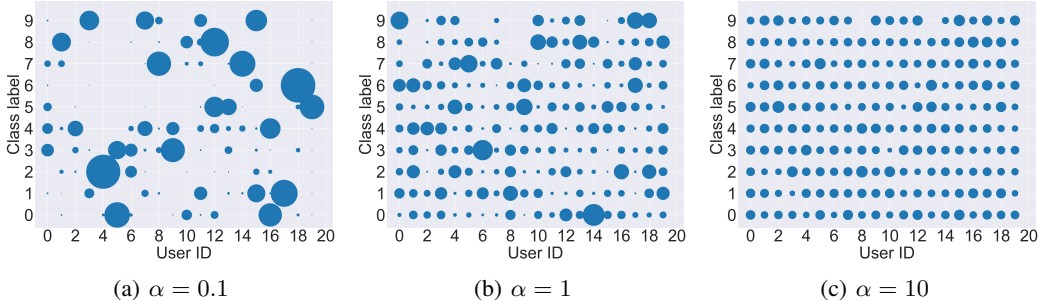

(a) $\alpha = 0.1$  (b) $\alpha = 1$  (c) $\alpha = 10$

Figure 7: Visulization of different levels of non-IID data distributions on the CIFAR-10 dataset, for different values of $\alpha$ of the Dirichlet distribution.

## C EXPERIMENT DETAILS

In this section, we present the details of our experiment implementation and used datasets. All our experiments were conducted on a system with 2 Intel® Xeon® Gold 6330 CPUs and 8 NVIDIA® GeForce RTX 3090 GPUs. We used three real datasets in this paper: CIFAR-10, CIFAR-100 (Krizhevsky, 2009), and Human Activity Recognition (HAR). For CIFAR-10 and CIFAR-100, we employed ResNet-18 as the model architecture, while for HAR, a CNN model with two convolutional layers was utilized. Table 4 describes the models and datasets, along with their total number of labels and data points used in our experiments.

Table 4: Description of models and datasets in our experiments.

| Model | Dataset | # of labels | # of data |
|---|---|---|---|
| ResNet-18 | CIFAR-10 | 10 | 60,000 |
| | CIFAR-100 | 100 | |
| CNN | HAR | 6 | 10.299 |

**Data Distribution:** We explore different levels of statistical heterogeneity (represented by varying values of parameters $\alpha$ and $N$, which are defined on page 8) across the network in two different ways. Note that both of these further yield unequal data volumes across the users. For each user, we randomly allocate 75% of the total data to the training dataset and the remaining 25% to the test dataset. The setting is summarized in Table 5.

Table 5: Statistics of imbalanced datasets associated with experiments in Table 1. Mean and Stdev list the mean and standard deviation of local data sizes.

| Dataset | levels of non-IID | Mean | Stdev |
|---|---|---|---|
| CIFAR-10 | $\alpha = 0.1$ | 3,000 | 1,434.73 |
| | $\alpha = 1$ | 3,000 | 563.97 |
| | $\alpha = 10$ | 3,000 | 233.21 |
| CIFAR-100 | $N = 20$ | 3,000 | 2,371.36 |
| | $N = 50$ | 3,000 | 4,140.07 |
| | $N = 100$ | 3,000 | 6,096.31 |
| HAR | - | 343.3 | 35.71 |

**Baselines:** We compare `pFedGT` against a variety of existing PFL methods such as `Ditto` (Li et al., 2021) and `pFedMe` (T Dinh et al., 2020), which learn personalized models that are maintained close to the global model by regularization. `APFL` (Deng et al., 2020) achieves personalization through *model interplolation*, i.e., a weighted average of the local and global *models*. `PerFedAvg` (Fallah et al., 2020) uses a meta-learning approach to learn local models based on each user task. In addition to evaluating our proposed `pFedGT` method, we also conducted comparisons with two established state-of-the-art representation learning methods, namely `FedRep` (Collins et al., 2021) and `FedBABU` (Oh et al., 2022). Both of these methods utilize a similar initialization technique, where the body of the neural network is initialized with the global model. However, the two methods differ in their initialization of the head layers. Specifically, `FedBABU` randomly initializes the head layers while `FedRep` initializes the head from the global model and then performs fine-tuning on the head while freezing the body. Additionally, we also conducted experiments on testing the global model on local test data through `FedAvg` (McMahan et al., 2017), as well as its fine-tuning (FT) method (Wang et al., 2019).

# D   ADDITIONAL EXPERIMENTS

In our main paper, the experimental results are conducted on the same fixed random seed. For completeness of presentation, we have done 3 runs of our experiments in CIFAR-10 dataset on different random seeds, and the results are shown in Figure 8. The results demonstrate pFedGT consistently yields better results than baselines (in fact the lower bar always lies above the upper bar of all baselines).

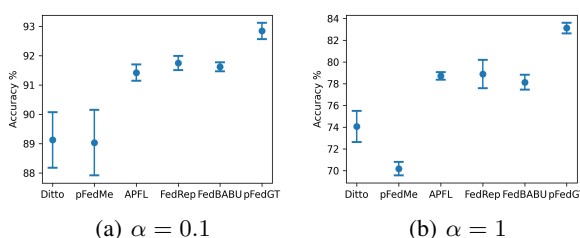

(a) $\alpha = 0.1$            (b) $\alpha = 1$

Figure 8: Results are shown over 3 runs on CIFAR-10 dataset.

**Effect of hyperparamter $\mu$.** We tune $\mu$ in a candidate set $\{0.01, 0.05, 0.1, 0.5\}$, then we fix $\mu = 0.05$ in all the experiments we conducted. The following table presents the ablation study for varying $\mu$ while fixing all other parameters (the level of heterogeneity is $\alpha = 0.1$ of Dirichlet distribution). The results indicate somehow that our algorithm is robust to a relatively small $\mu$ between 0.01 and 0.1.

Table 6: ablation study for $\mu$.

| Dataset | $\mu = 0.01$ | $\mu = 0.05$ | $\mu = 0.1$ | $\mu = 0.5$ |
|---|---|---|---|---|
| CIFAR-10 | 92.14% | 92.64% | 91.58% | 80.11% |
| CIFAR-100 | 60.10% | 59.84% | 58.70% | 23.91% |

**Ablation on local initialization.** We explore more on the local initialization of $w_i, c_i$ (line 1 of Alg. 2). Fig. 9 presents the results of these experiments. Notably, we observed that initializing local models with the server model $\theta$ provides an effective warm start for local personalization. Additionally, initializing users' local combined messages $c_i$ with the averaged value $c$ aids in stabilizing pFedGT, reducing oscillations, and achieving robust convergence. Consequently, these choices were adopted by default in the experiments for our proposed method.

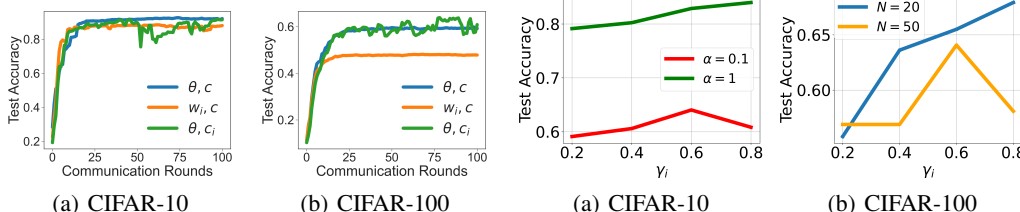

(a) CIFAR-10            (b) CIFAR-100            (a) CIFAR-10            (b) CIFAR-100

Figure 9: Initialization strategies involve using the server model $\theta$ or the previous local model $w_i$ for initializing the local model and either the global average $c$ or the prior local value $c_i$ for initializing $c_i$. Optimal results are achieved when $\theta$ and $c$ are used, as detailed in the main paper.

Figure 10: Accuracy comparison of a specific user for different values of personalization hyperparameter $\gamma_i$ in pFedGT while fixing the others. The optimal value of $\gamma_i$ varies depending on the specific data distribution, demonstrating the flexibility and adaptability of the proposed method in real-world scenarios.

**Personalization coefficient $\gamma_i$.** Through the use of the personalization coefficient $\gamma_i$, users can find a balance between the local and global information used in personalization. To investigate the effect of the personalization hyperparameter $\gamma_i$ on the performance of individual users, we conduct experiments by varying its value while keeping the other users fixed to $0.8$ (empirically set as the default value throughout the paper). Fig. 10 shows the performance comparison results for a specific user, highlighting that the optimal value of the personalization hyperparameter $\gamma_i$ varies with different data distributions, consistent with our initial design intention, as illustrated in Fig. 1. Besides, we conduct experiments on varying only for the **MAX** user and the **MIN** user (where the **MAX** user holds the largest data volume (5,994 data points), and the **MIN** user represents the opposite (1,164 data points)) while fixing the values (0.8) for all users in CIFAR-10 $\alpha = 0.1$ scenario. The results are shown in Table 7.

Table 7: Accuracies of the **MAX** and **MIN** user for different values of $\gamma_i$ in CIFAR-10 ($\alpha = 0.1$) while fixing the values (0.8) for all users.

| User | 0.1 | 0.2 | 0.3 | 0.4 | 0.5 | 0.6 | 0.7 | 0.8 | 0.9 |
|------|-----|-----|-----|-----|-----|-----|-----|-----|-----|
| **MAX** | 86.59% | 87.73% | 92.26% | 92.60% | **93.66%** | 92.46% | 93.46% | 93.00% | 93.33% |
| **MIN** | 71.82% | 86.23% | 86.94% | 90.03% | 88.66% | 89.00% | 89.35% | **90.38%** | 89.69% |

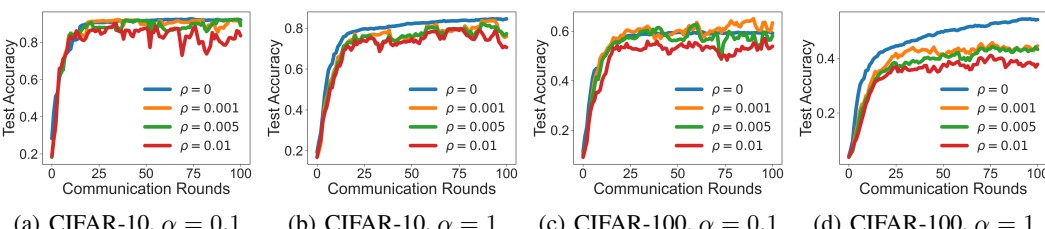

(a) CIFAR-10, $\alpha = 0.1$     (b) CIFAR-10, $\alpha = 1$     (c) CIFAR-100, $\alpha = 0.1$     (d) CIFAR-100, $\alpha = 1$

Figure 11: Our experiments indicate that when $\rho = 0$, the algorithm consistently achieves the best or second-best results across the scenarios we tested. In scenarios with a higher degree of non-IID data distribution, the performance gap does not significantly widen.

**Effect of hyperparamter $\rho$.** In our analysis, $\rho$ has to be sufficiently large to ensure the existence and uniqueness of the Nash equilibrium and the convergence of our algorithm. We would like to emphasize that the analysis is conservative, i.e., it provides a rigorous guarantee without excluding other choices for the operation of the algorithm (which we test experimentally). Subsequantly, we conducted experiments by using various values ($\{0, 0.001, 0.005, 0.01\}$) for the hyperparameter $\rho$. Fig. 11 shows that $\rho = 0$ (which we use by default in our experiments in the main paper) achieves the best or the second best results. Additionally, the performance gap in highly heterogeneous scenarios is not significant compared with that in more IID scenarios. This provides greater flexibility in adjusting the parameter $\rho$ during the implementation of pFedGT, beyond the lower bound established in our theorem.

