# OpenReview forum: "A Game-theoretic Approach to Personalized Federated Learning Based on Target Interpolation"
_ICLR.cc/2024/Conference — Submitted to ICLR 2024_

### Official Review · Reviewer_PnWa · 2023-10-26

**Soundness:** 3 good
**Presentation:** 3 good
**Contribution:** 2 fair
**Rating:** 5
**Confidence:** 3

**Summary:**

This paper addresses the issue of Personalized Federated Learning and proposes pFedGT, a method for personalized Federated Learning based on a Game-theoretic approach, that adopts a formulation termed “Target interpolation.” This paper conducts detailed experiments on the proposed algorithm, and the experimental results demonstrate that the algorithm achieves better performance on multiple datasets.

**Strengths:**

1. The experimental section of this paper is shown in detail, comparing the performance of the proposed algorithm with other algorithms on multiple datasets. The results indicate superior performance.

**Weaknesses:**

1. My major concern is the lack of novelty. The proposed idea of target interpolation in this papers seems to bear some resemblance to the concept of model interpolation presented in 'Three Approaches for Personalization with Applications to Federated Learning'(Mansour et al). From the algorithmic perspective, the essence of the algorithm proposed in this paper is still the introduction of a new regularization technique, unrelated to the game-theoretic approach.

2. Although this paper emphasizes that its algorithm is a game-theoretic approach, in reality, both the algorithm design and theoretical analysis lack the incorporation and analysis of game-theoretic principles. In fact, only a single sentence at the end of Section 3.1 briefly mentions the concept of Nash equilibrium and claims that each user iteratively solves the problem to achieve Nash equilibrium, which I doubt. I hope the authors can provide more theoretical and experimental analysis about game theory instead of merely mentioning the concept of Nash equilibrium.

3. The algorithm lacks protection for user model privacy. Unlike most federated learning approaches that update models by transmitting gradients in each round, the algorithm proposed in this paper transmits information c_i between the agent and server, where c_i is the gradient subtracted by the user's own model parameters. For most users, transmitting their own model parameters to the server is not acceptable compared to algorithms that only transmit gradients. (This is likely to happen when the user's model gradually converges, and c_i is approximately equal to μ w_i. Users who value model privacy are unlikely to accept this situation.)

**Questions:**

Please see the weakness.

---

> ### Author Response · Authors · 2023-11-21
>
> ### __Response to W1__
> Thank you very much for your questions. In traditional ML, training is carried out via loss minimization: a loss term is associated with each data point and the objective is to minimize the aggregated loss. In this regard, we believe that the "target interpolation" that we introduce in this paper is a natural extension for personalization: the impact of data is controlled by a parameter ($\gamma_i$ - which balances the contribution of local and global data mention also equation in the paper). In contrast, model interpolation is not interpretable. In view of the non-linearity and non-convexity of the task, model interpolation cannot be cast directly as a data balancing mechanism.
>
> ### __Response to W2__
> We thank you very much for your constructive suggestions. The reason is as follows.
>
> (1) Motivated by the limited data volume and the emphasis on personalized services, our proposed "target interpolation" ($\gamma_i f_i(w_i) + (1-\gamma_i) \frac{1}{m}\sum_{j = 1}^{m}f_j(w_i)$ problem (1), page 3) balances the contribution of local and global data for enabling personalized model learning. This formulation is for the objective of the learning task, and it is irrespective of the algorithm or a federated implementation.
>
> (2) In order to obtain an efficient federated implementation (in view of the fact that data cannot be communicated), we introduced a quadratic approximation ($f_j(w_i) \approx  f_j(w_j) + \nabla f_j(w_j)^{\top}(w_i - w_j)+\frac{\mu}{2}||w_i - w_j||^2$, page 4).
>
> (3) Afterwards, in our framework, we consider selfish users, i.e., they seek to achieve the best possible local models, whence a non-cooperative framework is adopted.
>
> Additionally, in our analysis, we define the response function of user $i$ as an operator $R_i(w) = \underset{w_i}{\text{arg min }} G_i(w_i;w_{-i})$ (Appendix B, page 14), which leverages information from all other users—specifically, their models $w_j$ and gradients $\nabla f_j(w_j)$—as input to generate its personalized model $w_i$. Consequently, we define $R(w)$ as the aggregation of all $R_i(w)$. Demonstrating that by choosing a sufficiently large $\rho$, the operator $R$ becomes contractive, we establish the existence and uniqueness of Nash equilibrium through the Banach fixed-point theorem.
>
> ### __Response to W3__
> Thank you very much for your comments. We want to explain that the privacy concerns are beyond this paper's topic. We would like to emphasize that our method is similar to all the FL methods that do not transmit any of the users' data. There are also FL methods that transmit model parameters [1][2], or transmit a combination of model parameters and dual variables [3][4]. In our case, we transmit a combination of model parameters and gradients. We would like to point out that when the user model converges, $c_i$ does not equal $\mu w_i$ (the gradient $f_i(w_i)$ is not equal to zero, instead, the partial gradient $\nabla_{w_i} G_i(w_i;w_{-i})$ equals zero).
>
>
>
> [1] McMahan, Brendan, et al. "Communication-efficient learning of deep networks from decentralized data." Artificial intelligence and statistics. PMLR, 2017.
>
> [2] T Dinh, Canh, Nguyen Tran, and Josh Nguyen. "Personalized federated learning with moreau envelopes." Advances in Neural Information Processing Systems 33 (2020).
>
> [3] Zhang, Xinwei, et al. "FedPD: A federated learning framework with adaptivity to non-iid data." IEEE Transactions on Signal Processing 69 (2021): 6055-6070.
>
> [4] Wang, Shuai, et al. "Beyond ADMM: a unified client-variance-reduced adaptive federated learning framework." Proceedings of the AAAI Conference on Artificial Intelligence. Vol. 37. No. 8. 2023.

---

### Official Review · Reviewer_5Hnv · 2023-10-29

**Soundness:** 2 fair
**Presentation:** 2 fair
**Contribution:** 2 fair
**Rating:** 5
**Confidence:** 4

**Summary:**

This paper studies the personalized federated learning (PFL) problem where local agents do not completely follow the global model and keep local models for their local demands. The authors claim they address the problem using the game-theoretic approach to model the PFL problem with a target interpolation (a linear combination of local objective and global objective), where the local deployed model parameters are considered the agents' strategies in the game. The authors then show that after adding a sufficiently strong L2 regularizer to the local objective, the PFL problem using the pFedGT algorithm will converge to a unique solution (Nash equilibrium).

**Strengths:**

The authors provides a complete story with problem formulation, algorithm pseudo code, theoretical convergence guarantee and numerical experiments showing the performance of pFedGT on the PFL problem defined in this paper.

**Weaknesses:**

1. The reason of formulating this PFL problem as a game theory problem is unclear. The updating dynamics is almost identical to FedAvg, where the authors replaced the "local loss" with "a linear combination of local loss and global loss". This change actually makes the agents more collaborative than strategically non-cooperative, and thus using a game theory framework for this problem is not providing any help in the intuition or the analysis.
2. There are places in the paper where the presentation can be significantly improved. For example,
(1). How is the heterogeneity level \alpha defined in Figure 1? Is it the same as the \alpha in Theorem 2? Can you provide intuition behind the strength of alpha and how the data will look like?
(2). On page 4, the discussion on the use of c is confusing until we check the pseudo code of Algorithm 2. The authors should definitely direct the readers to Algorithm 2 and provide more explanations on this
3. The claim that Assumption 1 is the only assumption is questionable, since the authors require a sufficiently strong regularizer to ensure strong convexity of the problem, which is restrictive in many applications beyond the Cifar classification. Moreover, under Lipschitz and strong convexity conditions, the uniqueness and convergence is very straightforward and there is no need for novel proof techniques.
4. How the aggregation interval and the partial participation schemes influence the convergence is not discussed in the theoretical results.
5. It is not easy to understand the difference between this work's setting and result from previous works, the authors should consider adding a table with each related works' setting, assumptions, solution existence and uniqueness, and convergence guarantee.

**Questions:**

1. Is the game theoretic framework a necessity? If so, why is that?
2. If the agents strategically change \gamma_i and only optimize the local loss, can your framework generalize to that and how may the results look like?

---

> ### Author Response · Authors · 2023-11-21
> **Response (1/2)**
>
> ### __Response to W1__
> We thank you very much for your constructive suggestions. The reason why we resort to a game theory is as follows.
>
> (1) Motivated by the limited data volume and the emphasis on personalized services, our proposed "target interpolation" ($\gamma_i f_i(w_i) + (1-\gamma_i) \frac{1}{m}\sum_{j = 1}^{m}f_j(w_i)$ problem (1), page 3) balances the contribution of local and global data for enabling personalized model learning. This formulation is for the objective of the learning task, and it is irrespective of the algorithm or a federated implementation.
>
> (2) In order to obtain an efficient federated implementation (in view of the fact that data cannot be communicated), we introduced a quadratic approximation ($f_j(w_i) \approx  f_j(w_j) + \nabla f_j(w_j)^{\top}(w_i - w_j)+\frac{\mu}{2}||w_i - w_j||^2$, page 4).
>
> (3) Afterwards, in our framework, we consider selfish users, i.e., they seek to achieve the best possible local models, whence a non-cooperative framework is adopted.
>
> Additionally, in our analysis, we define the response function of user $i$ as an operator $R_i(w) = \underset{w_i}{\text{arg min }} G_i(w_i;w_{-i})$ (Appendix B, page 14), which leverages information from all other users—specifically, their models $w_j$ and gradients $\nabla f_j(w_j)$—as input to generate its personalized model $w_i$. Consequently, we define $R(w)$ as the aggregation of all $R_i(w)$. Demonstrating that by choosing a sufficiently large $\rho$, the operator $R$ becomes contractive, we establish the existence and uniqueness of Nash equilibrium through the Banach fixed-point theorem.
>
> ### __Response to W2__
> Sorry for the confusion.
>
> (1): you are right, we mistakenly used the same symbol: the heterogeneity level $\alpha$ is different from that in Theorem 2. We have modified Theorem 2, and use notation $a$. We provide the visualization of data distribution of different values of $\alpha$ in Figure 7 on page 18.
>
> (2): Thank you very much for your constructive suggestions, we have modified the paper to give a clearer presentation based on your suggestions (Section 3, page 4).
>
> ### __Response to W3__
> Thank you very much for your comments. Assumption 1 is the only one needed to prove the existence and uniqueness of Nash equilibrium. The regularizer is not an assumption for our algorithm, this is a "hyperparameter". We completely agree with you that choosing a very large regularizer may not be a good option in a practical scenario. There are discrepancies between theory and practice, especially in neural network training. Our method still works without that regularizer, but we can not provide guarantees for such case.
> Regarding your last comment, making the local loss function strongly convex does not guarantee the existence, let alone the uniqueness, of a Nash equilibrium. First, the standard proof of the existence of Nash equilibrium requires a compact constrained set [1] (i.e., it does not apply to unconstrained optimization like in our setting). In addition, uniqueness is much harder to establish, and can only be done in specific cases [2]. Considering $f_i(w_i;w_{-i}) = 1/2||w_i-w_{(i+1)\text{ mod } n}||^2$ exhibits strong convexity concerning $w_i$. While it satisfies this property, it admits infinitely many Nash equilibria, such as when $w_1=w_2= ... =w_n$.
>
> ### __Response to W4__
> Thanks for your suggestions and we have modified Theorem 2 (Section 4, page 6). We analyzed Theorem 2 with a simple activation scheme of each user being active with some probability lower bounded by $p_{\text{min}}>0$ (strictly greater than zero is necessary otherwise there are users that never participate). At the end of this round, the server aggregates as exactly listed in Algorithm 1, step 7 on page 5. Under this randomization, we can establish our theorem. We also show Theorem 2 here for your convenience (Details are shown in Appendix B).
>
> __Theorem 2.__ Let Assumptions 1 and 2 hold. For any $\gamma_i\in (0,1)$, $\lambda = |S^t|/m$, $\rho>$max{$L,\underset{i}{\text{max }} L_{F_i}$}, and assume each client has a probability of being selected at each round that is lower bounded by a positive constant $p_{\mathrm{min}}>0$, then the following holds:
> \begin{equation}
>     \begin{split}
>         \mathbb{E}\big[|| w^{t} - w^\star||^2\big]\leq \frac{a^t}{p_{\mathrm{min}}} || w^{0}-w^\star ||^2 + \frac{1-a^t}{1-a}\sum_{i=1}^{m}\frac{(1+\xi)\varepsilon_i}{(\rho -L_{F_i})^2} , \nonumber
>     \end{split}
> \end{equation}
> where $a = 1-p_{min}(1-(1+\xi^{-1})a_1^2) \in (0,1)$, $\xi$ is an any constant that satisfies $\xi \geq a_1/(1-a_1)$, and $a_1=\underset{i}{\text{max }} \frac{(1-\gamma_i)(m-1)(\mu+L)}{(1-\gamma_i)(m-1)\mu-(\gamma_i(m-1)+1)L + m\rho}$, $L_{F_i}=\frac{(m-1)\gamma_i+1}{m}L + \frac{(1-\gamma_i)(m-1)}{m}\mu$.

---

> ### Author Response · Authors · 2023-11-21
> **Response (2/2)**
>
> ### __Response to W5__
> Thank you for your constructive suggestions, we have added the discussions on the comparison with baselines which show theoretical analysis in Appendix B.
>
> The Lipschitz continuity is the only assumption we need to prove the existence and uniqueness of the solution under suitable selection of the hyperparameter $\rho$ (as in Theorem 1). Furthermore, our algorithm converges linearly to that solution while the other PFL methods demonstrate a sublinear convergence rate ditto, pFedMe, PerFedAvg. There is no need for assumptions on the level of statistical heterogeneity (i.e., _bounded diversity_, $\frac{1}{m}\sum_{i=1}^{m}||\nabla f_i(w) - \nabla f(w)||\leq \sigma^2$, where $f(w):=\frac{1}{m}\sum_{i=1}^{m}f_i(w)$), which is commonly used in many PFL methods like Ditto, pFedMe, and PerFedAvg. In contrast to other papers, which first prove their global model converges, and then prove that their personalized models stay close to their global model, our method directly proves all the personalized models converge linearly (to a unique Nash equilibrium that depends only on the selection of $\gamma_i$ by the users). We summarize the comparison in the following table, where LC and Heterogeneity represent whether need to assume Lipschitz continuity and the level of statistical heterogeneity, respectively, and CR represents the convergence rate (note $a<1$).
>
> | Algorithm | LC | Heterogeneity | CR |
> |---|:--:|:---:|:---:|
> | ditto | &#10004; | &#10004; | $\mathcal{O}(\frac{1}{T})$ |
> | pFedMe | &#10004; | &#10004; | $\mathcal{O}(\frac{1}{T})$ |
> | PerFedAvg | &#10004; | &#10004; | $\mathcal{O}(\frac{1}{T})$ |
> | pFedGT | &#10004; | - | $\mathcal{O}(a^T), a\in (0,1)$ |
>
> [1] Başar, Tamer, and Geert Jan Olsder. Dynamic noncooperative game theory. Society for Industrial and Applied Mathematics, 1998.
>
> [2] Scutari, Gesualdo, et al. "Convex optimization, game theory, and variational inequality theory." IEEE Signal Processing Magazine 27.3 (2010): 35-49.
>
> ### __Response to Q1__
> We thank you very much for your questions. The reason why the game-theoretic framework is needed is as follows.
>
> (1) Motivated by the limited data volume and the emphasis on personalized services, our proposed "target interpolation" ($\gamma_i f_i(w_i) + (1-\gamma_i) \frac{1}{m}\sum_{j = 1}^{m}f_j(w_i)$ problem (1), page 3) balances the contribution of local and global data for enabling personalized model learning. This formulation is for the objective of the learning task, and it is irrespective of the algorithm or a federated implementation.
>
> (2) In order to obtain an efficient federated implementation (in view of the fact that data cannot be communicated), we introduced a quadratic approximation ($f_j(w_i) \approx  f_j(w_j) + \nabla f_j(w_j)^{\top}(w_i - w_j)+\frac{\mu}{2}||w_i - w_j||^2$, page 4).
>
> (3) Afterwards, in our framework, we consider selfish users, i.e., they seek to achieve the best possible local models, whence a non-cooperative framework is adopted.
>
> Additionally, in our analysis, we define the response function of user $i$ as an operator $R_i(w) = \underset{w_i}{\text{arg min }} G_i(w_i;w_{-i})$ (Appendix B, page 14), which leverages information from all other users—specifically, their models $w_j$ and gradients $\nabla f_j(w_j)$—as input to generate its personalized model $w_i$. Consequently, we define $R(w)$ as the aggregation of all $R_i(w)$. Demonstrating that by choosing a sufficiently large $\rho$, the operator $R$ becomes contractive, we establish the existence and uniqueness of Nash equilibrium through the Banach fixed-point theorem.
>
> ### __Response to Q2__
> Thank you very much for your question. We acknowledge that this is a limitation of our current experiments, but we would like to point out that even without a comprehensive means for tuning $\gamma_i$ at the user level, the simple setting of fixed and equal $\gamma_i$ manages to outperform baselines in a range of diverse settings (in terms of datasets and levels of heterogeneity).

---

### Official Review · Reviewer_Ha4r · 2023-10-31

**Soundness:** 2 fair
**Presentation:** 3 good
**Contribution:** 2 fair
**Rating:** 5
**Confidence:** 3

**Summary:**

This paper introduces an interesting personalized Federated Learning method. In this method, the local objective functions are modeled through a combination of the objective functions from all clients. Additionally, the authors present an approximation technique that allows for the estimation of objective functions from other clients without the necessity of transmitting local data. Experimental results demonstrate the effectiveness of the proposed approach.

**Strengths:**

1. Modeling clients' objective functions as a composite of individual clients' objective functions is promising.
2. The existence and uniqueness of a Nash equilibrium are provided.
3. The experiments demonstrate the proposed method is useful.

**Weaknesses:**

1. The hyper-parameter $\gamma$ is a crucial element controlling the strength of the objective functions of other clients. Nevertheless, the authors have not conducted adequate experiments to elucidate how algorithm performance varies with different values of $\gamma$.
2. In the case of Theorem 2, it appears that when $\gamma = 1$ (indicating no collaboration), the algorithms achieve the most favorable convergence results.
3. The formulation of Theorem 2 seems to address the convergence rate with only one local step, which suggests it may be more relevant to traditional distributed algorithms rather than federated learning algorithms.
4. Assumption 2 is not common in PFL. It would be better if more justification is provided.
5. The hyper-parameter $\rho$ plays a pivotal role in Theorem 2, and the theorems are only valid when $\rho \ge \max_{i} (L \cdot L_{F_i})$. However, the results in Figure 10 indicate that setting $\rho = 0$ consistently yields favorable results. While I understand the authors' choice to ensure strong-convexity by setting $\rho \ge \max_{i} (L \cdot L_{F_i})" for theoretical purposes, it introduces a significant disparity between theory and experimental outcomes.
6. Regarding Algorithm 1, the communication overhead seems heavy, as there is an additional $c^t$ that needs to be exchanged between server and clients, besides the model.

Minors:
1. It appears that optimizing the objective functions of other clients may impede the training convergence of the current client, as also corroborated by Theorem 2. However, the locally reported performance suggests that this impediment actually benefits the final performance. I am intrigued by this phenomenon and would appreciate more details from the authors regarding the construction of the training and test sets. Do the local training and test sets share the same distribution, or is the paper assessing generalization performance otherwise?
2. The approximation technique is not novel [1, 2]. Furthermore, it may be worthwhile to explore other methods for approximating Hessians, such as utilizing the Fisher Information Matrix or directly employing PyHessian.
3. It would be advantageous to include error bars in the experimental results for a more comprehensive presentation.

[1] Yin D, Farajtabar M, Li A. SOLA: Continual learning with second-order loss approximation[C]. 2020.
[2] Guo Y, Lin T, Tang X. A new analysis framework for federated learning on time-evolving heterogeneous data[J]. 2021.

**Questions:**

1. Could the authors give more explanations about Theorem 2?
2. Could the authors provide more discussions on the $\rho$ and $\gamma$?
3. Could the authors provide more details about the experiment settings? Additionally, the number of clients should be increased.

---

> ### Author Response · Authors · 2023-11-21
> **Response (1/2)**
>
> ### __Response to W1__
> We thank you very much for your constructive suggestions. Besides Figure 10 in Appendix D, we have conducted new experiments: we fixed the values for all users and varied only for the __MAX__ user and the __MIN__ user (where the __MAX__ user holds the largest data volume and the __MIN__ user represents the opposite). The following table represents the number of data points of the users.
>
> | CIFAR-10, $(\alpha =0.1)$ | MAX  | MIN  |
> |:------------------:|:------:|:------:|
> | # of data points | 5,994 | 1,164 |
>
> The results are as follows:
>
> | User| 0.1 | 0.2 | 0.3 | 0.4 | 0.5 | 0.6 | 0.7 | 0.8 | 0.9|
> |:---:|:--:|:--:|:--:|:--:|:--:|:--:|:--:|:--:|:--:|
> | __MAX__| 86.59 | 87.73|  92.26|  92.60| __93.66__ |  92.46|  93.46|  93.00|  93.33|
> | __MIN__| 71.82 | 86.23 | 86.94 | 90.03 | 88.66 | 89.00 | 89.35 | __90.38__ | 89.69 |
>
> The optimal value of the __MAX__ user is smaller than that of the __MIN__ user. This is due to the learning task of a user with a small number of data points being more biased to several classes.
>
> ### __Response to W2__
> Thank you very much for your comments. You are absolutely right: $\gamma = 1$ in our formulation means that each user does local training only. Thus, there is no game to play (in fact no communication is needed) which is also captured as a special case in Theorem 2 (page 6) (the first term in the right-hand side is zero). Therefore, the Nash equilibrium is the solution to the local training. It does not require any communication and, therefore no collaboration, which indicates the most favorable convergence results.
>
> ### __Response to W3__
> Sorry for causing this confusion. The proof is established between rounds $t$ and $t+1$, then by applying induction, we can get the exponential term. We have modified Theorem 2 in the paper and also included partial participation (Section 4, page 6).
>
> ### __Response to W4__
> We are sorry for causing this confusion. Assumption 2 is not an “assumption” needed to prove the convergence. We use it to represent the accuracy of the local training solutions. In our experiments, we capture the $\varepsilon_i$ by assigning the amount of local work to the users. The larger the local work is, the smaller $\varepsilon_i$ is. This is also been done in many papers like SCAFFOLD[1], FedProx[2], Ditto[3], pFedMe[4].
>
> ### __Response to W5__
> We appreciate your comments. You are absolutely right that $\rho$ has to be sufficiently large to ensure the existence and uniqueness of the Nash equilibrium and the convergence of our algorithm. We would like to emphasize that the analysis is conservative, i.e., it provides a rigorous guarantee without excluding other choices for the operation of the algorithm (which we test experimentally) (Figure 11, page 20).
>
> ### __Response to W6__
> Thank you very much for your question. In our framework, convergence can be guaranteed by only transmitting $c_i$. It is a trade-off between communication cost and convergence speed. Our method can establish the same communication cost per round as in the other FL methods, i.e., initialize local problems with local models $w_i$ (then there is no need for users to upload $w_i$, thus, in turn, they do not need to download the $\theta$). Given sufficient local work, initialization is non-important (this is due to we make the local objective strongly convex by choosing a large enough $\rho$ -- as a consequence of Lemma 2 in Appendix, page 13). We have done ablations on local initialization choices (Figure 9, Appendix D), which shows that initializing with $\theta$ yields better accuracy when given the same amount of local work.
>
> [1] Karimireddy, Sai Praneeth, et al. "Scaffold: Stochastic controlled averaging for federated learning." International conference on machine learning. PMLR, 2020.
>
> [2] Li, Tian, et al. "Federated optimization in heterogeneous networks." Proceedings of Machine learning and systems 2 (2020): 429-450.
>
> [3] Li, Tian, et al. "Ditto: Fair and robust federated learning through personalization." International Conference on Machine Learning. PMLR, 2021.
>
> [4] T Dinh, Canh, Nguyen Tran, and Josh Nguyen. "Personalized federated learning with moreau envelopes." Advances in Neural Information Processing Systems 33 (2020).

---

> ### Author Response · Authors · 2023-11-21
> **Response (2/2)**
>
> ### __Response to Minor 1__
> Thank you very much for your question. The reasons are twofold:
>
> (1) Users contend with limited data, and our proposed "target interpolation" technique facilitates the indirect incorporation of all other users' data.
>
> (2) In our experimental setup, the training/testing datasets share the same labels, which contributes to heightened accuracy.
>
> ### __Response to Minor 2__
> We thank you for your constructive suggestions. First, it is commonly used in optimization. Second (this is the most important reason), if we use a matrix to approximate the Hessians ($\frac{1}{2}||w_i-w_j||^2_{\tilde{H}_j}$), this will lead to a gradient $\tilde{H}_j(w_i-w_j)$. However, during the local updating of user $i$, while $w_i$ changes, $\tilde{H}_j$ remains fixed. Computing the term $\tilde{H}_j w_i$ necessitates transmitting the matrix, posing a substantial burden on communication overhead.
>
> ### __Response to Minor 3__
> Thank you very much for your constructive suggestions. We have plotted the error bar in Appendix D. We kindly refer you to Figure 8, page 19.
>
> ### __Response to Q1__
> Thank you very much for your suggestions, we have modified Theorem 2 along with its remark based on your and the other reviewer's comments (Section 4, page 6). Let me show you here for your convenience.
>
> Our theorems provide the guarantees of the effectiveness of our method under a large enough $\rho$. However, the analysis is conservative, in that it does not exclude the other choices of $\rho$. Our analysis requires no assumption on the level of statistical heterogeneity (i.e., bounded diversity -- the variance of local gradients to global gradient is bounded, in contrast to several methods in the FL literature, i.e., pFedMe, PerFedAvg, Ditto. Theorem 2 is established under any selection of $\gamma_i \in (0,1)$, our problem formulation guarantees the existence uniqueness of Nash equilibrium. Further, our algorithm converges linearly to the equilibrium. Moreover, the convergence rate $a=\mathcal{O}(1)$ as a function of the system size $m$, which supports the scalability of the proposed algorithm. Theorem 2 is established with a simple activation scheme of each user is active with probability lower bound by $p_{\mathrm{min}}>0$. In practice, this assumption is necessary otherwise the user will never participate.
>
> ### __Response to Q2__
> We thank you very much for your questions. As discussed in our previous response to your inquiry,
>
> (1) $\rho$ has to be sufficiently large to ensure the existence and uniqueness of the Nash equilibrium and the convergence of our algorithm. We would like to emphasize that the analysis is conservative, i.e., it provides a rigorous guarantee without excluding other choices for the operation of the algorithm (which we test experimentally) (Figure 11, page 20).
>
> (2) The parameters $\gamma_i$ pertain to the problem formulation itself. They can be freely chosen by the users (in fact, we note in brief that the quadratic regularization was adopted so as not to impose any constraint in the choices of $\gamma_i\in (0,1)$ in Theorems 1,2).
>
> ### __Response to Q3__
> Thank you very much for your suggestions, we have added more details about the experiment settings in Appendix C (page 18). We have done ablations on the system scale (Table 2, page 9). The results demonstrate that our algorithm consistently yields the best performance.

---

### Official Review · Reviewer_jCxY · 2023-11-08

**Soundness:** 3 good
**Presentation:** 3 good
**Contribution:** 2 fair
**Rating:** 5
**Confidence:** 3

**Summary:**

This work proposed a new personalized federated learning model based on a weighted average of local and global loss, and further approximate it into a game formulation. The proposed model attains Nash equilibrium, the corresponding algorithm attains linear convergence. Extensive numerical experiments further showcased the superiority of the algorithm.

**Strengths:**

1. New model for PFL
2. The proposed algorithm outperforms existing works in the experiments.

**Weaknesses:**

1. The additional introduced regularization term $\frac{\rho}{2}||w_i||^2$ term is weakly motivated, as far as I understand, this term is more like for theory convenience, which makes the function to be strongly convex, so the Nash existence and linear convergence are expected to some extent. I may think the idea is a bit similar to that of FedProx [1].
2. Lots of hyperparameters concerning the function objectives ($\mu, L, \gamma_i$) are required compared to classical algorithms, which weaken the practical significance.
3. Mismatch between theory and practice. You mentioned in the experiments you choose $\rho=0$ works (and it even outperforms over other choices), while the $\rho$ is required to be larger than $L$ in the theory. Such mismatch has not been highlighted and thoroughly discussed.
4. In fact that also raises my concern about whether the authors need to resort to a game theory background for the paper. If the additional regularization term is only for theory convenience to attain Nash, while it seems to be a bit unnecessary in the experiment, I think the algorithm and storyline of the paper can be revised. With a nonconvex objective only, the proposed algorithm should still be able to converge to stationarity.

[1] Li, Tian, et al. "Federated optimization in heterogeneous networks." Proceedings of Machine learning and systems 2 (2020): 429-450.

**Questions:**

See above

---

> ### Author Response · Authors · 2023-11-21
>
> ### __Response to W1__
> Thank you very much for your question. You are absolutely right that this is necessary for the analysis: $\rho$ has to be sufficiently large to ensure the existence and uniqueness of the Nash equilibrium and the convergence of our method. We would like to emphasize that the analysis is conservative, i.e., it provides a rigorous guarantee without excluding other choices for the operation of the algorithm (which we test experimentally). You are also right that quadratic regularization is a common technique. Nevertheless, we would like to underline that strong convexity of the local objectives does not guarantee immediately existence or uniqueness of NE. The reason is twofold: a) classical analysis heavily relies on optimization on a compact set (while we adopt unconstrained optimization that allows for the optimality condition ($\nabla_{w_i} G_i(w_i;w_{-i})=0$ in page 4) and b) strong convexity does not imply uniqueness of NE (for example, consider $f_i(w_i;w_{-i}) = 1/2||w_i-w_{(i+1)\text{ mod } n}||^2$, which admits infinitely many Nash equilibriums, $w_1=w_2= ... =w_n$).
>
> ### __Response to W2__
> Thanks for your comments. We agree with you about the importance of hyperparameter tuning in a practical scenario.
>
> (1) $L$ is not a parameter of the algorithm but it is only used for analysis (through the standard Assumption 1) and leads to a lower bound for $\rho$. As we mentioned above, this is a conservative requirement to provide rigorous guarantees, and it is not crucial to estimate $L$ which, for a neural network, is far for straightforward task [1][2].
>
> (2) Regarding $\mu$, in our experiments, we tune it in a candidate set {0.01, 0.05, 0.1, 0.5}, then we fix $\mu=0.05$ in all the experiments we conducted. The following table presents the best results conducted on various $\mu$ values while fixing all the others (the level of heterogeneity is $\alpha =0.1$ of Dirichlet distribution). The results show that our algorithm is robust for a relatively small $\mu$ between 0.01 and 0.05.
>
> |Dataset| $\mu=0.01$ | $\mu=0.05$ | $\mu=0.1$ | $\mu=0.5$ |
> |:---------:|:---:|:---:|:--:|:--:|
> | CIFAR-10 | 92.14% | __92.64%__ | 91.58% | 80.11% |
> | CIFAR-100| __60.10%__ | 59.84% | 58.70% | 23.91% |
>
> (3) The parameters $\gamma_i$ pertain to the problem formulation itself and can be freely chosen by the users (in fact, we note in brief that the quadratic regularization was adopted so as not to impose any constraint in the choices of $\gamma_i\in (0,1)$ in Theorems 1,2).
>
> ### __Response to W3__
> Thank you very much for your comments. The reason why we add a regularization term is to provide theoretical guarantees. You are absolutely right that there is a discrepancy between theory and practice, and this is reminiscent of all analyses of optimization methods. The reason is in the proof we use bounds to reach the desired, and there is no way to assess how 'loose' these bounds are in a practical setting. For this reason, we resort to experimentation (Figure 11 on page 20).
>
> ### __Response to W4__
> We thank you very much for your constructive suggestions. The reason why we resort to a game theory is as follows.
>
> (1) Motivated by the limited data volume and the emphasis on personalized services, our proposed "target interpolation" ($\gamma_i f_i(w_i) + (1-\gamma_i) \frac{1}{m}\sum_{j = 1}^{m}f_j(w_i)$ problem (1), page 3) balances the contribution of local and global data for enabling personalized model learning. This formulation is for the objective of the learning task, and it is irrespective of the algorithm or a federated implementation.
>
> (2) To obtain an efficient federated implementation (in view of the fact that data cannot be communicated), we introduced a quadratic approximation ($f_j(w_i) \approx  f_j(w_j) + \nabla f_j(w_j)^{\top}(w_i - w_j)+\frac{\mu}{2}||w_i - w_j||^2$, page 4).
>
> (3) Afterwards, in our framework, we consider selfish users, i.e., they seek to achieve the best possible local models, whence a non-cooperative framework is adopted.
>
> Additionally, in our analysis, we define the response function of user $i$ as an operator $R_i(w) = \underset{w_i}{\text{arg min }} G_i(w_i;w_{-i})$ (Appendix B, page 14), which leverages information from all other users—specifically, their models $w_j$ and gradients $\nabla f_j(w_j)$—as input to generate its personalized model $w_i$. Consequently, we define $R(w)$ to be the concatenation of all $R_i(w)$. Demonstrating that by choosing a sufficiently large $\rho$, the operator $R$ is contractive. Thus we establish the existence and uniqueness of Nash equilibrium through the Banach fixed-point theorem.
>
> [1] Fazlyab, Mahyar, et al. "Efficient and accurate estimation of lipschitz constants for deep neural networks." Advances in Neural Information Processing Systems 32 (2019).
>
> [2] Zhang, Bohang, et al. "Rethinking lipschitz neural networks and certified robustness: A boolean function perspective." Advances in Neural Information Processing Systems 35 (2022).

---

> > ### Comment · Reviewer_jCxY · 2023-11-23
> > **Thank you**
> >
> > Thank you for the response.
> >
> > I agree with the authors that the analysis may be conservative, and there is a discrepancy between theory and practice in the community. But with such discrepancy, it will be harder to rationalize the motivation, also to pinpoint the contribution and significance of the proposed model and algorithm. I also went through other reviews and I concur with other reviewers and the work can be further enhanced. I will maintain my score here. Thank you.

---

### Author Response · Authors · 2023-11-21
**General response to all reviewers**

Dear Reviewers,

We would like to thank all reviewers for taking the time to review our paper, and for your valuable comments and suggestions that helped a lot to improve both the content and the presentation of our paper. We have addressed all your comments in the revised paper (edits are shown in red) and accompanying individual responses. In the following, we provide some key changes in our revised paper along with the page number.

(1) We provide additional explanations about the non-cooperative game formulation in Section 1, page 2, and Section 3, page 4;

(2) We have modified Theorem 2 and added more explanations and discussion on our theoretical analysis. Specifically, (i) a new version of Theorem 2 that considers partial participation along with the discussion on the hyperparameters in Remark 1 (Section 4, page 6) and Appendix B, page 17; (ii) a comparison table and discussions with the baselines that have a complete analysis in Table 3 (Appendix B, page 17);

(3) A supplementary description of the experimental setting in Appendix C, page 18;

(4) in Appendix D (pages 19-20), we added the error bar plot (Figure 8 on page 19), and ablations on the choice of $\mu$ (Table 6 on page 19) and $\gamma_i$ (Table 7 on page 20), as well as more explanations for $\rho$ (page 20).

Finally, we would like to thank you again for your valuable time and constructive comments.

Sincerely,
Authors of Paper 4673

---

### Meta-Review · Area_Chair_rAuG · 2023-12-06

**Metareview:**

This paper proposes an approach for federated learning that interpolates between a local loss and a global loss which is not entirely accessible to the end-user. The paper finds that that the method is able to converge to a Nash equilibrium. The reviewers viewed the modeling approach, theoretical development, and experimental tests as strengths. However, they identified some weaknesses concerning the connection between the theory and implementation and sensitivity analysis of hyperparameters. The authors engaged in the response period and even conducted additional experiments to support the claims of the paper. The reviewers appreciated the responses, but elected to keep their ratings below the acceptance threshold. A tighter connection between the theory and implementation would be needed to elevate the paper to acceptance.

**Justification For Why Not Higher Score:**

The reviewers remained concerned about the coupling between theory and implementation. The paper is much improved from the initial submission, but remains below the bar.

**Justification For Why Not Lower Score:**

N/A

---

### Decision · Program_Chairs · 2024-01-16

Reject